# Molecular determinants regulating selective binding of autophagy adapters and receptors to ATG8 proteins

Martina Wirth[1], Wenxin Zhang [1,2,5], Minoo Razi[1,5], Lynet Nyoni[1,2], Dhira Joshi[3], Nicola O'Reilly [3], Terje Johansen [4], Sharon A. Tooze [1] & Stéphane Mouilleron [2]

Autophagy is an essential recycling and quality control pathway. Mammalian ATG8 proteins drive autophagosome formation and selective removal of protein aggregates and organelles by recruiting autophagy receptors and adaptors that contain a LC3-interacting region (LIR) motif. LIR motifs can be highly selective for ATG8 subfamily proteins (LC3s/GABARAPs), however the molecular determinants regulating these selective interactions remain elusive. Here we show that residues within the core LIR motif and adjacent C-terminal region as well as ATG8 subfamily-specific residues in the LIR docking site are critical for binding of receptors and adaptors to GABARAPs. Moreover, rendering GABARAP more LC3B-like impairs autophagy receptor degradation. Modulating LIR binding specificity of the centriolar satellite protein PCM1, implicated in autophagy and centrosomal function, alters its dynamics in cells. Our data provides new mechanistic insight into how selective binding of LIR motifs to GABARAPs is achieved, and elucidate the overlapping and distinct functions of ATG8 subfamily proteins.

[1] Molecular Cell Biology of Autophagy, The Francis Crick Institute, 1 Midland Road, London NW1 1AT, UK. [2] Structural Biology Science Technology Platforms, The Francis Crick Institute, 1 Midland Road, London NW1 1AT, UK. [3] Peptide Chemistry Science Technology Platforms, The Francis Crick Institute, 1 Midland Road, London NW1 1AT, UK. [4] Molecular Cancer Research Group, Department of Medical Biology, University of Tromsø – The Arctic University of Norway, 9037 Tromsø, Norway. [5] These authors contributed equally: Wenxin Zhang, Minoo Razi. Correspondence and requests for materials should be addressed to M.W. (email: martina.wirth@crick.ac.uk) or to S.A.T. (email: sharon.tooze@crick.ac.uk) or to S.M. (email: stephane.mouilleron@crick.ac.uk)

Autophagy is an essential stress survival pathway transferring cytoplasmic material in autophagosomes to lysosomes for degradation, thereby restoring nutrients and molecular building blocks[1]. Autophagy also ensures cellular health by selectively removing toxic macromolecules, damaged organelles, or intracellular pathogens[2,3]. Accordingly, deregulation of autophagy has been implicated in a broad range of diseases, including cancer, neurodegenerative disorders, and infection.

The signaling and machinery proteins mediating autophagosome formation are highly conserved from yeast to human. In mammals, the ULK protein kinase complex initiates autophagosome formation by phosphorylating and activating the ATG14-Beclin1-phosphatidylinositol 3-phosphate (PI3P) kinase complex[4,5]. Production of PI3P at autophagosome formation sites on the endoplasmic reticulum (ER) (omegasomes) recruits PI3P-binding effectors, DFCP1 and WIPI proteins[6]. WIPI2b is essential for the recruitment of the ATG12-5-16L1 complex[7], which mediates in a ubiquitin-like conjugation reaction covalent attachment of cytosolic ATG8 proteins to phosphatidylethanolamine on the autophagic membrane[8,9].

Mammalian ATG8 proteins comprise two subfamilies, namely LC3s (LC3A, LC3B, and LC3C) and GABARAPs (GABARAP, GABARAP-L1, and GABARAP-L2)[10]. ATG8 proteins promote autophagosome formation, elongation, and closure, as well as fusion with lysosomes[11–13]. In selective autophagy, ATG8 proteins (on the inner membrane) facilitate engulfment of cargo in autophagosomes by directly binding to autophagy receptors, such as p62[14], NDP52[15], or NBR1[16]. However, a growing number of ATG8 interactors are autophagy adaptors, which are not degraded by autophagy and fulfilling diverse functions ranging from regulation of autophagosome formation (e.g., ULK complex[17,18]) and fusion with the lysosome (e.g., PLEKHM1[19]) to autophagosome transport (e.g., FYCO1[20]).

Both autophagy receptors and adaptors contain an ATG8-interacting motif, more commonly known as LC3-interacting region (LIR) motif, which is recognized by ATG8 proteins[21–23]. The canonical LIR motif is a small $\Theta_0$-$X_1$-$X_2$-$\Gamma_3$ motif, where $\Theta$ represents an aromatic residue (W/F/Y) and $\Gamma$ an aliphatic residue (L/V/I) (whose side chains bind to hydrophobic pocket 1 (HP1) and HP2 of the LIR docking site (LDS), respectively) and X represents any amino acid (aa). The N-terminal region directly preceding the core LIR motif often harbors acidic or phosphorylated residues that stabilize ATG8 binding through electrostatic interactions[21–23].

Despite overlapping functions there is growing evidence for functional differences between LC3s and GABARAPs[11–13,24]. Many LIR-containing proteins exhibit high selectivity for specific ATG8 proteins; however, the underlying mechanisms regulating binding selectivity are still poorly understood. A recent study focusing on residues in the core LIR motif, defined a GABARAP interaction motif (GIM), $\Theta$-[V/I]-$X_2$-V, where valine or isoleucine in position $X_1$ and valine in position $\Gamma_3$ promotes interaction with GABARAPs[25]. However, no LC3 subfamily-specific interaction motif has been identified and not all GABARAP-specific LIRs possess a GIM, suggesting that there may be additional mechanisms regulating selective binding to LC3 and GABARAP subfamily proteins.

In this study, we examined the molecular determinants that regulate selective binding of the centriolar satellite (CS) marker protein PCM1 (pericentriolar material 1) and the ULK1 complex to GABARAPs. We identified key residues within the LIR motif and immediately C-terminal to the core LIR motif, as well as non-conserved, subfamily-specific LDS residues in ATG8 proteins, which confer binding specificity towards GABARAPs. Altering the binding specificity of the PCM1 LIR motif changes PCM1

dynamics in cells. Furthermore, selective degradation of the autophagy receptor NDP52 is impaired by rendering GABARAP more LC3B-like. Our data provide new mechanistic detail on how LIR binding specificity is achieved and will help elucidating the diverse biological functions of ATG8 proteins.

## Results

**PCM1 binds to GABARAP via a C terminally extended LIR motif.** We recently showed that the CS protein PCM1 binds directly to GABARAP through a canonical LIR motif regulating GABARAP-specific autophagosome formation and GABARAP stability[26]. PCM1 is a large scaffolding protein, coordinating assembly of CS, which are electron-dense granules surrounding the centrosome and implicated in centrosome assembly and ciliogenesis[27]. We identified one main LIR motif (PCM1[1955–1958]:FVKV) in the C-terminus of PCM1[26]. Mutation of this LIR reduces GABARAP binding by more than 80%.

To elucidate the molecular basis of the PCM1-GABARAP interaction, we determined the crystal structure of the PCM1[1951–1964] LIR motif bound to GABARAP. The structure was solved in two different space groups $P2_12_12_1$ (Fig. 1a, b and Supplementary Fig. 1b) and $P4_3$ (Fig. 1c and Supplementary Fig. 1a, c) at a resolution of 1.55 and 1.86 Å (Table 1), respectively. Both structures displayed identical canonical LIR interactions comprising PCM1 hydrophobic residues F1955 ($\Theta_0$) and V1958 ($\Gamma_3$) deeply bound to HP1 and HP2, as well as three hydrogen bonds formed between the main chains of PCM1 LIR residues V1956 ($X_1$) and V1958 ($\Gamma_3$) and the main chains of GABARAP residues K48[GAB] and L50[GAB]. On top of these canonical LIR interactions additional specific contacts were observed in both PCM1:GABARAP structures. Within the core LIR motif, V1956[PCM1] in position $X_1$ formed a hydrophobic interaction with Y49[GAB] of HP2. Notably, the adjacent C-terminal region engaged in multiple interactions: (1) a salt bridge between E1959[PCM1] ($X_4$) and R28[GAB], (2) a large hydrophobic contact between L1961[PCM1] ($X_6$) and L55[GAB], F62[GAB], L63[GAB], and (3) two hydrogen bonds between the side chain of Q59[GAB] and the main chain of P1962[PCM1] ($X_7$), as well as K1964[PCM1] ($X_9$) through a water molecule (Fig. 1a–c and Supplementary Fig. 1a). This proline residue at position $X_7$ was also critical for GABARAP binding to a mutational peptide array of the PCM1 LIR motif[26].

Most of the residues interacting with the C-terminal region of the PCM1 LIR are only conserved among GABARAP subfamily proteins (R28[GAB], L55[GAB], Q59[GAB], and F62[GAB]) and LC3C (L64[LC3C] and Q68[LC3C]) (Fig. 1d). Thus, these residues may be critical for stabilizing association of the C-terminal region and mediating selective binding to the PCM1 LIR motif.

PCM1 binds strongly to GABARAP, GABARAP-L1, GABARAP-L2, and LC3C, and weakly to LC3A and LC3B[26] (Fig. 1e), which is the same binding behavior as reported for the ULK1 complex members ULK1, ATG13, and FIP200[17,18,25] (Fig. 1e). The PCM1 LIR motif is similar to the human ULK1 LIR motif[17,18,28] (Fig. 1f). PCM1, ULK1, and ATG13 LIR sequences contain GIMs[25] and interestingly the C-terminal regions of the PCM1, ATG13, ULK1, and FIP200 LIR motifs also exhibit proline residues at position $X_{6–8}$.

To better understand the selective ATG8 binding of these proteins, we used bio-layer interferometry (BLI) to determine the binding affinities ($K_d$ values) of the individual LIR motifs to all recombinant ATG8 proteins (Fig. 1g).

The ULK1 LIR bound the strongest to GABARAP and GABARAP-L1 with $K_d$s of 50 and 48 nM, respectively, followed by the ATG13 LIR ($K_d$s of 0.59 and 0.53 μM), which also showed a relatively strong affinity for LC3A ($K_d$ of 4.1 μM). Both PCM1

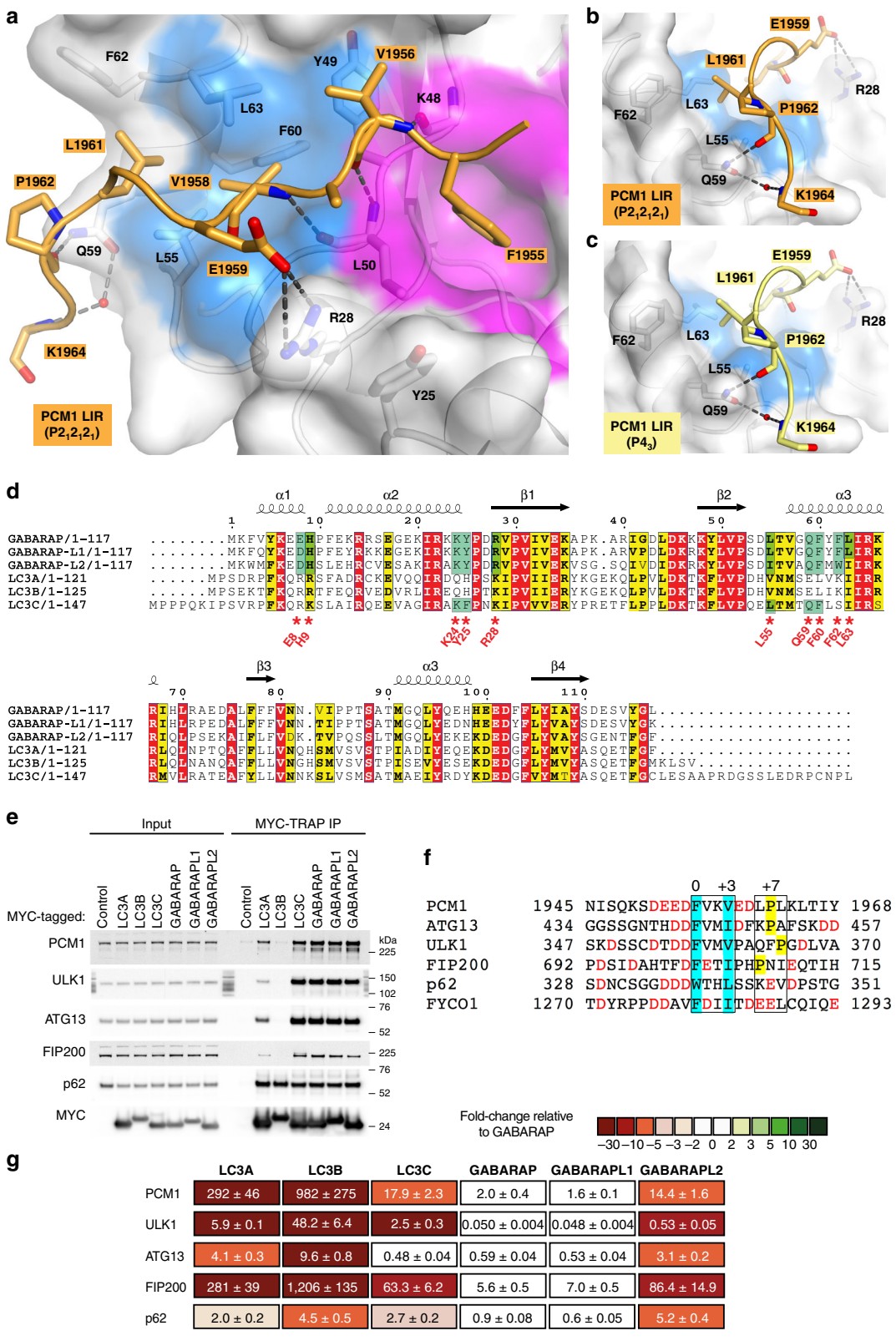

and FIP200 LIRs bound weakly to GABARAP/GABARAP-L1 with $K_d$s of 2.0/1.6 and 5.6/7.0 μM, respectively. The LIR motif of the autophagy receptor p62 (SQSTM1) bound strongly to all six ATG8 proteins. The p62 LIR affinity to LC3B (4.5 μM) is similar to the affinity (3.2 μM) determined by isothermal calorimetry using a C terminally extended p62 LIR peptide[29]. p62 binding to

LC3B (4.5 μM) and GABARAP-L2 (5.2 μM) was five-fold weaker than to GABARAP (0.9 μM); however, these differences in binding are not detected in MYC-Trap immunoprecipitation (IP) experiments (Fig. 1e), where p62, due to PB1 domain-mediated polymerization, binds strongly to all mammalian ATG8 proteins[14]. The binding specificities of ULK1, ATG13, FIP200,

**Fig. 1** Pericentriolar material 1 (PCM1) binds to GABARAP via a C terminally extended LC3-interacting region (LIR) motif. **a** Structure of PCM1[1951-1964] LIR bound to GABARAP ($P2_12_12_1$). The PCM1 LIR sequence is shown in orange cartoon with interacting residues shown as sticks. GABARAP is displayed in white cartoon and transparent surface with hydrophobic pocket 1 and 2 colored in pink and blue surfaces, respectively. **b, c** Close-up view of Q59[GAB] interaction with main chain carbonyl residues of P1962[PCM1] and K1964[PCM1] in two different crystal forms (**b**: $P2_12_12_1$; **c**: $P4_3$). **d** Sequence alignment of human ATG8-family orthologs using ESPript[61]. Identical (red) and similar residues (yellow) are boxed. Red asterisks and green boxes indicate non-conserved residues between LC3A/B and LC3C/GABARAPs further analyzed in this study. **e** MYC-TRAP immunoprecipitation (IP) of HEK293A cells expressing MYC-ATG8 constructs and immunoblot with indicated proteins. **f** Sequence alignment of LIR peptides. The core LIR motif is boxed and aromatic and hydrophobic residues in position $\Theta_0$ and $\Gamma_3$ depicted in blue. Residues in position $X_{6-8}$ are boxed to highlight proline residues (yellow). Acidic residues are shown in red. **g** Affinities ($K_d$ values) of LIR peptides to human ATG8 proteins determined by bio-layer interferometry (BLI). Color code indicates fold changes relative to wild-type (WT) GABARAP (data are mean ± s.d., $n = 2-4$)

**Table 1 Data collection and refinement statistics**

|  | PCM1:GABARAP | PCM1:GABARAP | ATG13:GABARAP | ULK1:GABARAP |
|---|---|---|---|---|
| PDB ID | 6HYL | 6HYM | 6HYN | 6HYO |
| Resolution range | 54.32–1.55 (1.61–1.55) | 45.59–1.86 (1.92–1.86) | 41.29–1.14 (1.18–1.14) | 41.12–1.07 (1.10–1.07) |
| Space group | $P2_12_12_1$ | $P4_3$ | $I23$ | $I23$ |
| Unit cell | 53.6 65.9 95.7 90 90 90 | 80.8 80.8 55.1 90 90 90 | 101.1 101.1 101.1 90 90 90 | 100.7 100.7 100.7 90 90 90 |
| Total reflections | 257,013 (24,635) | 140,914 (13,154) | 1,120,671 (76,985) | 1,385,490 (108,563) |
| Unique reflections | 49,150 (4840) | 30,131 (3045) | 62,488 (6252) | 74,520 (7428) |
| Multiplicity | 5.2 (5.1) | 4.7 (4.3) | 17.9 (12.3) | 18.6 (14.6) |
| Completeness (%) | 99.0 (95.7) | 99.5 (97.7) | 99.9 (99.9) | 99.9 (100.0) |
| Mean $I$/sigma ($I$) | 11.1 (1.2) | 19.7 (1.3) | 23.7 (1.0) | 23.9 (1.6) |
| Wilson $B$-factor | 24.3 | 41.5 | 15.7 | 14.0 |
| $R$-merge | 0.05 (0.97) | 0.02 (0.78) | 0.04 (1.82) | 0.04 (1.65) |
| $R$-meas | 0.06 (1.08) | 0.02 (0.89) | 0.04 (1.90) | 0.05 (1.71) |
| $R$-pim | 0.02 (0.47) | 0.01 (0.42) | 0.01 (0.54) | 0.01 (0.44) |
| CC1/2 | 0.998 (0.69) | 1 (0.65) | 1 (0.55) | 1 (0.62) |
| CC* | 1 (0.90) | 1 (0.88) | 1 (0.84) | 1 (0.87) |
| Reflections used in refinement | 48,794 (4680) | 30,008 (2978) | 62,465 (6252) | 74,519 (7428) |
| Reflections used for $R$-free | 2397 (261) | 1524 (192) | 3077 (326) | 3593 (400) |
| $R$-work | 0.24 (0.50) | 0.18 (0.37) | 0.22 (0.39) | 0.14 (0.23) |
| $R$-free | 0.28 (0.57) | 0.21 (0.40) | 0.23 (0.41) | 0.16 (0.24) |
| Number of non-hydrogen atoms | 2443 | 2366 | 1390 | 1414 |
| Macromolecules | 2209 | 2146 | 1219 | 1191 |
| Ligands |  | 28 |  | 52 |
| Solvent | 234 | 192 | 171 | 171 |
| Protein residues | 260 | 264 | 132 | 131 |
| RMS (bonds) | 0.00 | 0.02 | 0.01 | 0.01 |
| RMS (angles) | 0.52 | 1.24 | 1.30 | 1.39 |
| Ramachandran favored (%) | 99.6 | 99.6 | 99.2 | 98.5 |
| Ramachandran allowed (%) | 0.4 | 0.4 | 0.8 | 1.5 |
| Ramachandran outliers (%) | 0 | 0 | 0 | 0 |
| Average $B$-factor | 41.4 | 53.0 | 27.5 | 20.8 |
| Macromolecules | 41.1 | 52.6 | 26.3 | 17.9 |
| Ligands |  | 63.3 |  | 39.7 |
| Solvent | 44.1 | 56.4 | 36.2 | 35.6 |

and PCM1 LIR peptides were consistent with IP experiments (Fig. 1e). The strongest binding was measured for GABARAP and GABARAP-L1, followed by GABARAP-L2 and LC3C. LC3B was the weakest interactor with all four LIR peptides.

**ATG13 and ULK1 bind GABARAP via extended LIR motifs**. Structural analyses of many LIR motifs demonstrated that N-terminal acidic residues directly preceding the core LIR motif participate in ATG8 binding[21–23]. However, a role of the adjacent C-terminal region in stabilizing ATG8 binding, as reported, for example, for FYCO1[30–32], ALFY[33], and AnkG/AnkB[34], seems rather the exception.

Our affinity measurements showed that the LIR motifs of ATG13 and ULK1 exhibited very strong binding affinities for GABARAP. In order to find out whether their C-terminal regions are contributing to GABARAP binding in a similar way as observed for PCM1, we determined the crystal structure of ATG13[441–454] and ULK1[354–366] LIR motifs bound to GABARAP

at a resolution of 1.14 and 1.07 Å, respectively (Table 1 and Supplementary Fig. 1d, e). The structure of ATG13 bound to LC3A, LC3B, and LC3C was determined in a previous study[35]. However, residues C-terminal of the ATG13 core LIR motif were not included and ATG13 LIR structures in complex with GABARAP subfamily members have not been reported.

Both ATG13 and ULK1 structures (Fig. 2 and Supplementary Fig. 2) were solved near atomic resolution, which allowed the modeling of two alternate conformations for the ATG13 LIR in its C terminal moiety (Fig. 2a–c and Supplementary Fig. 2a, b). The ATG13 and ULK1 LIR motifs interacted with GABARAP in a very similar way (Supplementary Fig. 2c, d). In the N terminus, the main chain of the aspartate in position $X_{-2}$ (D442[ATG13] and D355[ULK1]) formed a hydrogen bond with K48[GAB] from HP1. Within the core LIR sequence ($\Theta_0$-$X_1$-$X_2$-$\Gamma_3$) the interactions were completely identical: (1) the canonical LIR interactions previously described for PCM1 structure, (2) in position $X_1$, a valine (V445[ATG13], V358[ULK1]) was packed against the aromatic

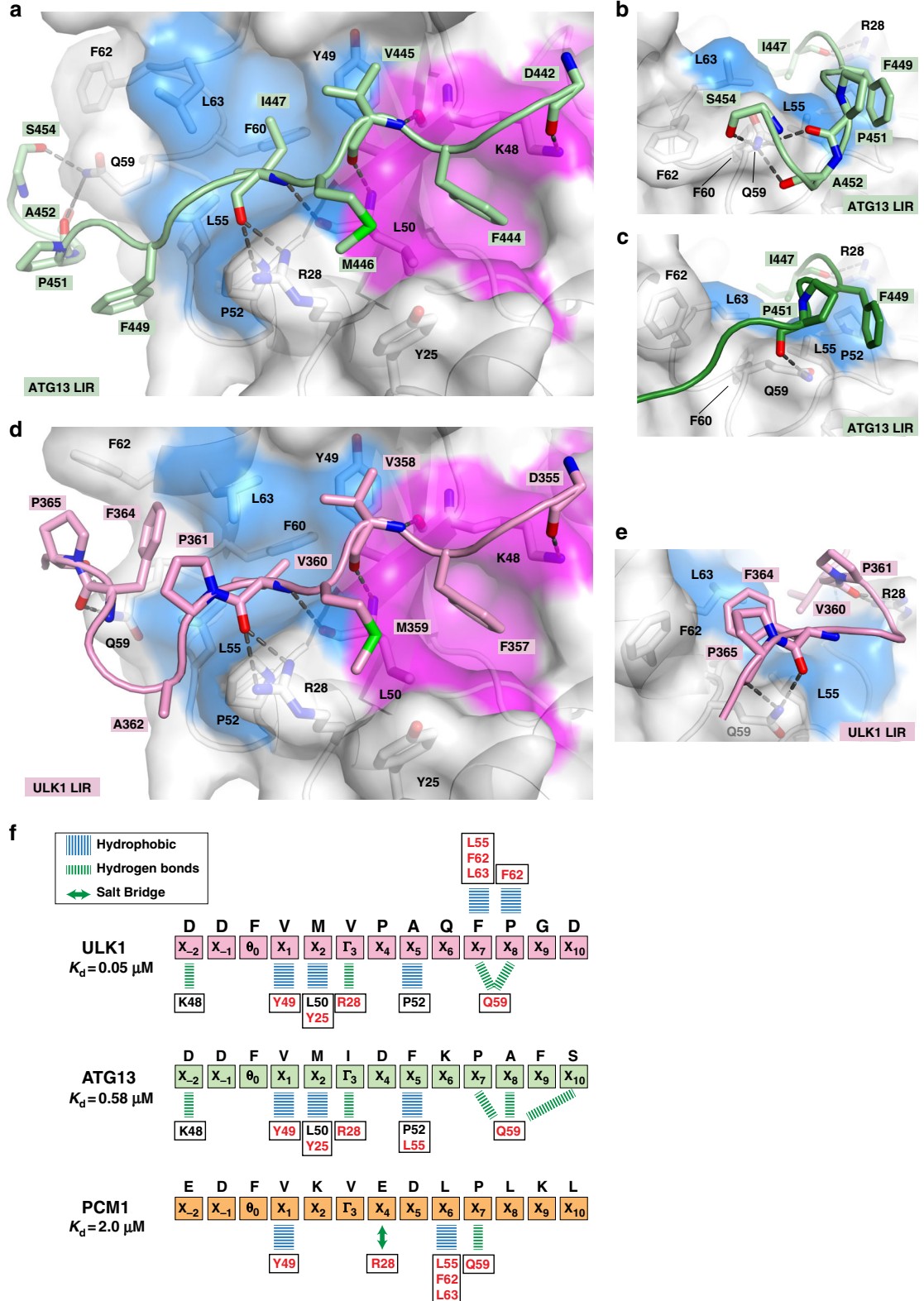

**Fig. 2** ATG13 and ULK1 bind GABARAP via C terminally extended LC3-interacting region (LIR) motifs. **a** Structure of the ATG13[441–454] LIR bound to GABARAP in alternate conformation A. The ATG13 LIR sequence is shown in light green cartoon with interacting residues depicted as sticks. GABARAP is displayed in white cartoon and transparent surface with hydrophobic pocket 1 (pink) and 2 (blue). **b, c** Close-up view of Q59[GAB] forming hydrogen bonds with main chain carbonyls of ATG13 LIR residues in alternate conformation A (light green) and B (dark green), respectively. **d** Structure of ULK1[354–366] LIR-GABARAP complex. The ULK1 LIR sequence is shown in pink cartoon with interacting residues depicted as sticks. **e** Close-up view of interactions between GABARAP residues, Q59[GAB], F62[GAB], and L63[GAB], with ULK1 LIR residues in position X[7,8] (F364[ULK1], P365[ULK1]). **f** Schematic overview of non-canonical LIR interactions observed in the structures of GABARAP bound to ULK1 (pink), ATG13 (green), and PCM1 (orange) LIR motifs. Blue lines indicate hydrophobic interactions, green lines hydrogen bonds, and green double arrow salt bridges. Strictly conserved residues are boxed and displayed in black; non-conserved residues are boxed and displayed in red

ring of $Y49^{GAB}$, (3) in position $X_2$, a methionine was engaged in a large surface of hydrophobic interactions with $L50^{GAB}/Y25^{GAB}$, (4) in position $\Gamma_3$, the carbonyl of the hydrophobic residue formed two hydrogen bonds with the guanidinium group of $R28^{GAB}$. The C-terminal region of ATG13 and ULK1 LIR motifs showed similar interactions. $F364^{ULK1}$ in position $X_7$ formed an extended hydrophobic interaction with $L55^{GAB}$, $F62^{GAB}$, and $L63^{GAB}$ (similarly to $L1961^{PCM1}$, Supplementary Fig. 2d). This interaction was further stabilized by $P365^{ULK1}$ in position $X_8$, which was packed against $F62^{GAB}$ side chain and formed hydrophobic contacts with $F364^{ULK1}$. Thus $F364^{ULK1}$ was sandwiched between $P365^{ULK1}$ and $L55^{GAB}/F62^{GAB}/\ L63^{GAB}$. In position $X_5$, ATG13 has a phenylalanine residue, $F449^{ATG13}$, which engaged in more limited hydrophobic interactions with $L55^{GAB}$ and $P52^{GAB}$ from HP2 (Fig. 2a–c and Supplementary Fig. 2a, b). $A361^{ULK1}$, the corresponding $X_5$ residue in the ULK1 LIR, made hydrophobic contact only with $P52^{GAB}$ (Fig. 2d). Finally, in both LIRs main chain carbonyls in position $X_7$, $X_8$, and $X_{10}$ ($P451^{ATG13}$, $A452^{ATG13}$, $S454^{ATG13}$ and $F364^{ULK1}$, $P353^{ULK1}$) formed hydrogen bonds with $Q59^{GAB}$ side chain (Fig. 2b–e and Supplementary Fig. 2c).

Overall, the structures of GABARAP bound to ATG13 and ULK1 LIRs exhibited similar interactions like the PCM1 LIR-GABARAP complex (Fig. 2f and Supplementary Fig. 2c, d). Within the core LIR motif, a Valine in position $X_1$ formed hydrophobic contact with $Y49^{GAB}$. However, in contrast to $M446^{ATG13}$ and $M359^{ULK1}$, the side chain of $K1957^{PCM1}$ in position $X_2$ was partially disordered and the aliphatic part formed only limited hydrophobic contact with GABARAP.

In the C-terminal region, hydrophobic residues in position $X_{5-7}$ interacted with the edge of the HP2, which is extended in GABARAP subfamily proteins by the presence of hydrophobic and aromatic residues ($L55^{GAB/GABL1}/I55^{GABL2}$ and $F62^{GAB/GABL1}/W62^{GABL2}$), but not in LC3 subfamily proteins, which have fewer hydrophobic, and even charged, residues ($V58^{LC3A/B}/L64^{LC3C}$ and $K65^{LC3A/B}/S71^{LC3C}$) in the corresponding positions (Fig. 1d). Most strikingly, hydrogen bonds and hydrophobic contacts of residues C-terminal to the core LIR motifs (positions $X_{4-10}$) seem to be critical in mediating strong association with GABARAP and involve residues, which are non-conserved between LC3 and GABARAP subfamily proteins. Hence, not only residues within the core LIR motif but also those C-terminal may confer binding selectivity to different ATG8 orthologs.

**C-terminal region critical for ULK1 LIR binding specificity.** The C-terminal regions of the PCM1 LIR and ULK complex LIR motifs are highly conserved (Supplementary Fig. 3a). To examine whether the C-terminal region may regulate selective binding to ATG8 proteins, we performed a peptide array analysis as well as BLI affinity measurements using truncated peptides of the ULK1 LIR motif (Supplementary Fig. 3b, c). Either six, eight, or all ten amino acids were deleted from the N or C terminus adjacent to the core LIR motif. Deletion of the whole sequence N-terminal to the core motif led to complete loss of binding (Supplementary Fig 3b) and a more than 200-fold decrease in affinity to all six ATG8 proteins (Supplementary Fig. 3c) underscoring the importance of acidic residues in positions $X_{-1}$ and $X_{-4}$ for the binding affinity[21]. Strikingly, removal of the whole C terminus of the ULK1 LIR motif switched on binding to LC3A and LC3B (Supplementary Fig. 3b, c), suggesting that the proline residue and/or the alanine residue in position $X_{4,5}$ ($P361^{ULK1}$ and $A362^{ULK1}$) may inhibit LC3A and LC3B binding to the ULK1 LIR motif. Using this peptide array approach, removal of the C-terminal region of the PCM1 LIR motif did not reveal any changes in binding specificity.

To further elucidate the role of $P361^{ULK1}$ and $A362^{ULK1}$ we analyzed the binding specificity of GST-GABARAP, GST-LC3B, and GST-LC3A to the ULK1 LIR motif using a mutational peptide array scan (Fig. 3a, b and Supplementary Fig. 3d). Every position within the 24-mer ULK1 LIR peptide was mutated to all alternative aa, revealing the importance of each individual residue within the LIR motif. The binding pattern of GST-LC3B (Fig. 3b) and GST-LC3A (Supplementary Fig. 3d) showed that residues in position $X_4$, but also position $X_{-3}$, $X_2$ seem unfavorable for LC3A/B binding. Mutation of $P361^{ULK1}$ (position $X_4$) to almost any other aa increased LC3A/B binding. To confirm the importance of residues in position $X_{-3}$, $X_2$, and $X_4$, we performed a peptide array analysis using peptides that carry single point mutations (Fig. 3c), which strongly increased LC3B binding in the mutational peptide array scan (Fig. 3b). Whereas $T354E^{ULK1}$ resulted in a slight increase, both $M359I^{ULK1}$ and $P361D^{ULK1}$ point mutations strongly increased the binding of all three LC3 subfamily members (Fig. 3c). $T354E/M359I/P361D^{ULK1}$ substitutions lead to the strongest increase in binding of LC3 proteins. Consistently, HA-ULK1 T354E/M359I/P361D and endogenous ULK complex members ATG13 and FIP200 show substantial increased binding to LC3B in GST pull-down experiments (Fig. 3d).

This underscores the importance of position $X_{-3}$, $X_2$, and $X_4$ within the ULK1 LIR motif in regulating selective binding to GABARAP over LC3 subfamily proteins. Moreover, our data show that the C-terminal region of the ULK1 LIR is involved in directing binding specificity towards GABARAP subfamily proteins and LC3C.

**Position $X_2$ in the PCM1 core LIR motif inhibits LC3 binding.** Which residues in the PCM1 LIR regulate binding specificity? Our structural analysis showed that the region C-terminal to the PCM1 core LIR motif contributes to stabilizing GABARAP interaction. However, in contrast to the ULK1 LIR motif, deletion of the C-terminal region of PCM1 did not increase interaction with LC3 subfamily members. To determine the role of the C-terminal region and/or the other regions of the PCM1 LIR motif in mediating selective binding to GABARAP proteins, we tested chimera peptide sequences of PCM1 and FYCO1 LIR motifs for GST-ATG8 protein binding in a peptide array experiment (Fig. 4a). We chose FYCO1 because it preferentially interacts with LC3A and LC3B and several structural studies showed that the C-terminal region stabilizes LC3 binding[30–32].

Replacing the PCM1 LIR C-terminus by the corresponding FYCO1 C-terminus did not increase LC3A and LC3B binding. However, introduction of the FYCO1 core LIR into the PCM1 LIR motif strongly increased LC3 subfamily binding. This interaction was further enhanced when both the FYCO1 core LIR and FYCO1 C-terminal region were present. On the other hand, the PCM1 core LIR motif strongly inhibited binding of the FYCO1 chimera LIR motif to both LC3 and GABARAP subfamily members. In addition, the PCM1 C-terminus strongly decreased binding to LC3A and LC3B.

The PCM1 core LIR motif was most critical for inhibiting LC3 subfamily binding. Consequently, we mutated individual residues to the corresponding residue present in the FYCO1 core LIR motif and determined ATG8 binding affinities (Fig. 4c). As observed before, introduction of the FYCO1 core LIR motif ($X_{1,2}$ and $\Gamma_3$) specifically activated LC3 subfamily binding. Whereas mutation of the GIM motif residues ($X_1$ and $\Gamma_3$) had opposing or only mild effects, mutation of the lysine in position $X_2$ ($K1957^{PCM1}$) dramatically increased both LC3 subfamily (LC3B: 95-fold; LC3A/LC3C: 75- and 47-fold, respectively) and GABARAP subfamily (~12-fold) binding. Mutations of residues N-terminal ($X_{-2, -1}$) and C-terminal ($X_4$, $X_7$) did not enhance

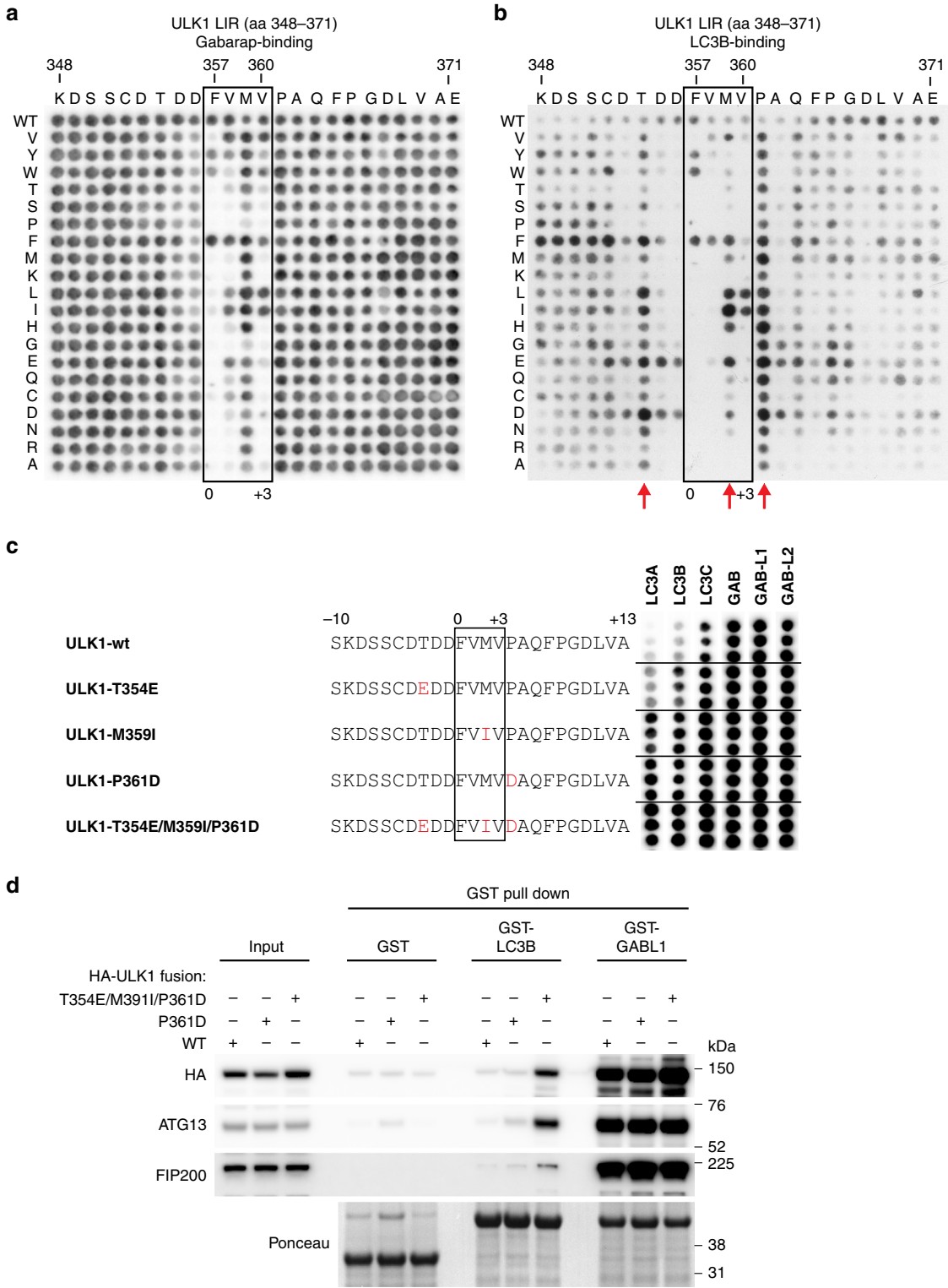

**Fig. 3** The C-terminal region is critical for ULK1 LC3-interacting region (LIR) binding specificity. **a**, **b** Mutational peptide array of 24-mer ULK1 peptide covering LIR motif incubated with GST-GABARAP (**a**) or GST-LC3B (**b**) and immunoblotted with anti-GST. Each amino acid position was substituted for every other amino acid. **c** Twenty four-mer peptide array analysis of ULK1 LIR peptides containing point mutations incubated with indicated GST-ATG8 protein and immunoblotted with anti-GST. Each peptide is spotted in triplicates. Mutated residues are highlighted in red. **d** GST pulldown of HEK293A cells expressing indicated HA-ULK1 constructs and immunoblot with indicated antibodies

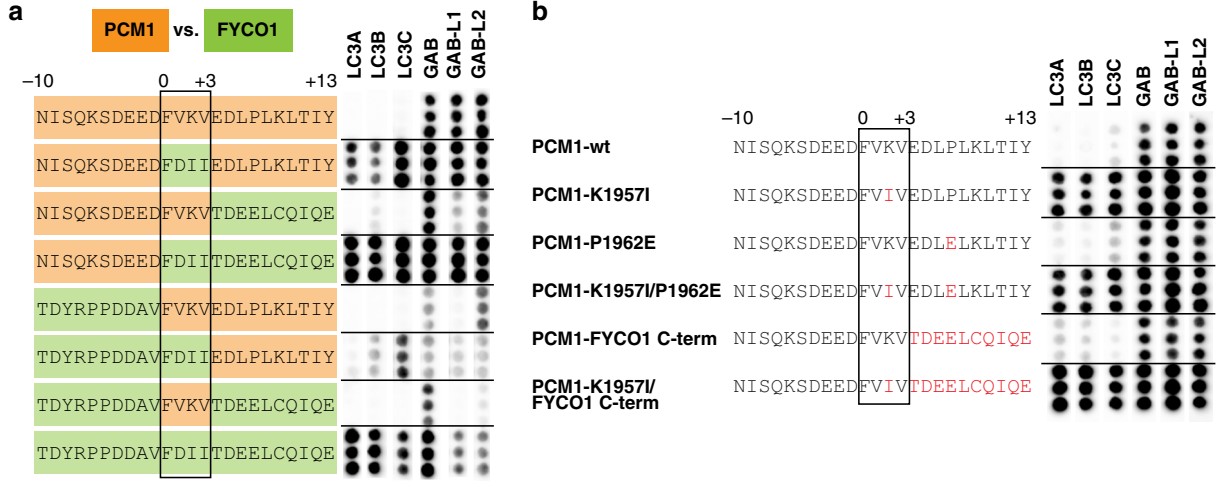

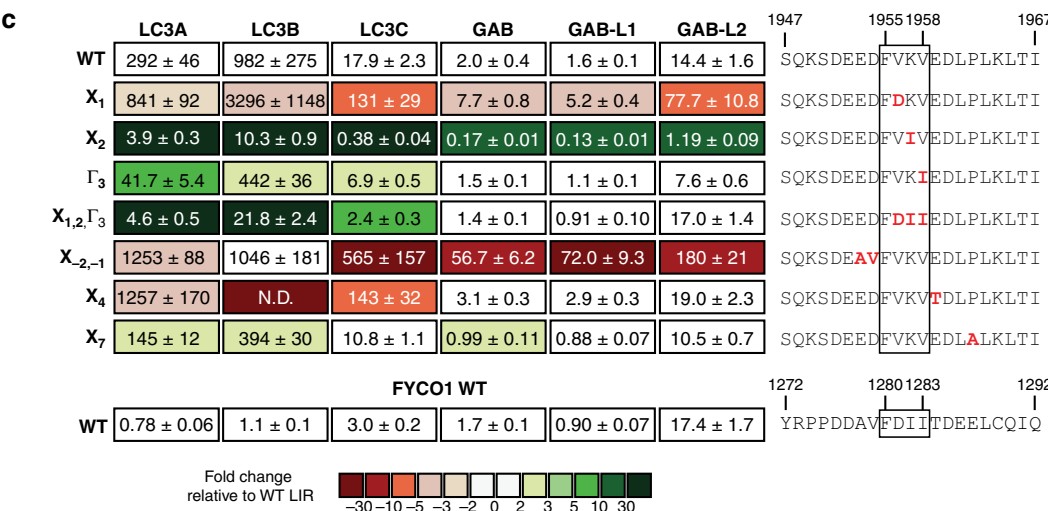

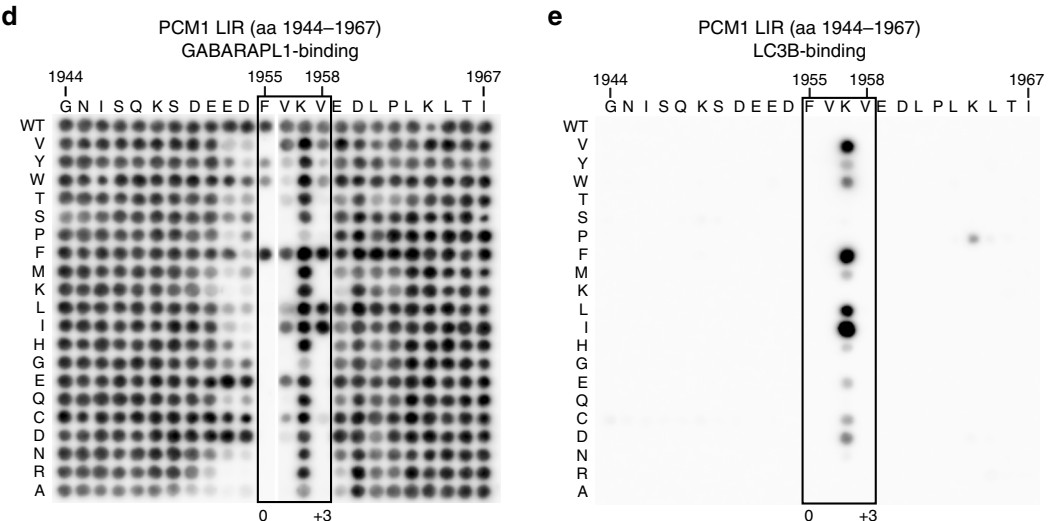

LC3 subfamily binding, indicating that LC3B binding was primarily regulated by K1957I[PCM1].

A mutational peptide array scan (Fig. 4d, e) further corroborated that the lysine residue in position $X_2$ alone is sufficient in inhibiting LC3B binding. Single mutation of K1957[PCM1], especially to aliphatic aa (isoleucine, leucine, and valine) and phenylalanine, strongly increased GST-LC3B association (Fig. 4e). Although GST-GABARAP[26] and GST-GABARAP-L1 (Fig. 4d) binding was slightly enhanced by mutations of position $X_2$, overall, GABARAP proteins seem to be able to

**Fig. 4** Position $X_2$ in the pericentriolar material 1 (PCM1) core LC3-interacting region (LIR) motif inhibits LC3 binding. **a** Twenty four-mer peptide array analysis of PCM1-FYCO1 LIR chimera sequences incubated with indicated GST-ATG8 protein and immunoblotted with anti-GST. Each peptide is spotted in triplicates. PCM1 and FYCO1 peptide sequence are highlighted in orange and green, respectively. **b** Twenty four-mer peptide array analysis of PCM1 LIR containing point mutations incubated with GST-ATG8 protein and immunoblotted with anti-GST. Each peptide is represented in triplicates. Mutated residues are highlighted in red. **c** Affinities ($K_d$ values) of LIR peptides to ATG8 proteins determined by bio-layer interferometry (BLI). Color code indicates fold changes relative to $K_d$ value of PCM1 WT LIR peptide binding to the corresponding ATG8 protein. (data are mean ± s.d., $n = 2$). **d**, **e** Mutational peptide array of 24-mer PCM1 peptide covering LIR motif incubated with GST-GABARAP (**d**) or GST-LC3B (**e**) and immunoblot. Each amino acid position was substituted for every other amino acid

tolerate almost any aa in that position of the PCM1 LIR motif, except for proline and glycine[26] (Fig. 4d). This intolerance for proline or glycine substitutions at $X_2$ was previously shown for ULK1 and ATG13 LIRs[18].

Consequently, introduction of a lysine residue at $X_2$ reduced more LC3 than GABARAP subfamily binding to the ATG4B, FUNDC1, and FKBP8 LIR motifs (Supplementary Fig. 4d–f).

In the PCM1 LIR peptide array scan, GST-GABARAP binding was affected by mutation of the proline residue in position $X_7$[26], but neither affinity measurements (Fig. 4c) nor peptide array experiments (Fig. 4b–d) substantiated an important role of P1962[PCM1] in GABARAP subfamily binding. A point mutational peptide array clearly confirmed the regulatory function of K1957[PCM1] (Fig. 4b). Moreover, in the presence of K1957I[PCM1] the FYCO1 C terminus also contributed to ATG8 subfamily binding by further increasing binding.

Conforming to the PCM1 and ULK1 LIR motifs, positions $X_2$ and $X_4$ regulated LC3 subfamily binding to the FIP200 LIR motif. Both T704I[FIP200] and P706D[FIP200] mutation allowed LC3 subfamily binding but also enhanced interaction with GABARAPs (Supplementary Fig. 4a, b, g). Notably, mutation of position $X_2$ of the FIP200 LIR motif (T704I[FIP200]) increases LC3 binding by 19- to 42-fold and GABARAP binding by 11- to 25-fold (Supplementary Fig. 4c).

In the case of the FYCO1 LIR motif (Supplementary Fig. 4h), introduction of a lysine in $X_2$, a proline in $X_4$, or the whole C terminus of either ULK1 or PCM1 weakened LC3A and LC3B interaction.

In summary, position $X_2$ and $X_{4-13}$ of the C-terminal region of LIRs were critical in regulating subfamily-specific binding to ATG8 proteins.

**Increased LC3 binding alters PCM1 dynamics in cells**. The K1957I[PCM1] mutation markedly changed the specificity of the LIR motif towards binding all ATG8s. Does this mutation affect PCM1 function in cells? GST pull-down experiments demonstrated that full-length GFP-PCM1 K1957I interacted strongly with both LC3 and GABARAP subfamily proteins (Fig. 5a). On the other hand, GFP-PCM1 wild type (WT) selectively interacted with GABARAPs and, to a lesser extent, with LC3C. GFP-PCM1 D1954A/F1955A/V1958A (PCM1 3xAla)[26] was deficient in ATG8 binding. The PCM1 LIR motif is required for recruiting peripheral PCM1-positive CS to sites of autophagosome formation, but not for PCM1 localization to the centrosome[26].

First, we tested whether the K1957I[PCM1] mutation affected PCM1 localization to the centrosome (centriole and pericentriolar material (PCM) marked by γ-tubulin) in starved HEK293A cells. Loss of ATG8 binding resulted in a significant increase of centrosomal GFP-PCM1 (3xAla) relative to total GFP-PCM1 (Fig. 5b and Supplementary Fig. 5a, b), i.e. more PCM1 was retained on centrosomes. On the other hand, increased ATG8 protein binding (PCM1 K1957I) lead to a small reduction in the amount of GFP-PCM1 on the centrosome compared to WT. More strikingly, in the cell periphery PCM1 K1957I formed large cytosolic granules, which were partially positive for pericentrin and SSX2IP (Supplementary Fig. 5a, b), known PCM1 interactors

and CS and PCM markers[36]. This suggests that increased ATG8 binding may alter CS integrity.

In response to starvation a subset of PCM1-positive CS colocalize with a range of autophagy markers, including LC3[26]. Switching on LC3 subfamily binding and enhancing ATG8 binding overall may recruit more PCM1-positive CS to forming autophagosomes and increase PCM1 colocalization with LC3. Indeed, PCM1 K1957I colocalized significantly more with LC3 than PCM1 WT and PCM1 3xAla in starved cells (Fig. 5c and Supplementary Fig. 5c). Moreover, inhibition of lysosomal degradation by bafilomycin A1 (BafA1) enhanced LC3 colocalization of PCM1 K1957I compared to PCM1 WT. Colocalization between PCM1 3xAla and LC3 was not altered by BafA1 treatment. The responsiveness to BafA1 treatment suggested that activation of LC3 binding may target PCM1 K1957I to autophagosomes for lysosomal degradation. Hence, we also quantified PCM1 colocalization with the lysosomal marker LAMP1 (Fig. 5d and Supplementary Fig. 5c). PCM1 WT colocalization with LAMP1 was slightly increased by BafA1 treatment, PCM1 3xAla was unchanged, while PCM1 K1957I showed a significant increase. To resolve whether PCM1 K1957I is recruited into autophagosomes or associates with the outer membrane of autolysosomes, we expressed tandem fluorescence (EYFP-mCherry)-tagged PCM1 in HEK293A cells. The acidic pH of the lysosome quenches the EYFP signal faster than mCherry[37] and red-only puncta indicate autolysosomal PCM1. Compared to EYFP-mCherry PCM1 WT and 3xAla, PCM1 K1957I showed a significant increase in red-only puncta (Fig. 5e). To exclude any effects arising from increased affinity to GABARAPs, a PCM1 V1956D/K1957I/V1958I mutant (PCM1 FDII), where the PCM1 core LIR motif is replaced by the FYCO1 core LIR motif, was generated. BLI experiments showed the PCM1 FDII mutant had strongly increased affinities to LC3s without altering affinities to GABARAPs (Fig. 4c). Both PCM1 K1957I and PCM1 FDII formed red-only puncta in starved HEK293A cells (Fig. 5e), and colocalization between EYFP and mCherry was significantly reduced compared to PCM1 WT (Fig. 5f). Western blot analysis did not reveal any lysosomal turnover of overexpressed EYFP-mCherry-PCM1 K1957I and FDII (Fig. 5g), suggesting that only a small fraction of PCM1 is degraded by autophagy. This is in line with previous findings showing that PCM1 is not degraded by autophagy[26,38].

Interestingly, both enhanced binding of all ATG8s (PCM1 K1957I) or only LC3s (PCM1 FDII) resulted in formation of these larger PCM1 granules (Fig. 5e), suggesting that increased LC3 binding alters the dynamics of PCM1 in cells.

**Non-conserved GABARAP residues key to selective LIR binding**. To fully understand the molecular mechanism underlying selective binding of ATG8 proteins to PCM1 and the ULK1 complex, we also addressed the role of the ATG8 protein sequence. Our structural analyses revealed several key residues within the LIR-GABARAP interface that were not conserved between the LC3 and GABARAP subfamilies (Figs. 1d, 2f). For instance, Q59[GAB] is only conserved among GABARAPs and LC3C. Both LC3A and LC3B have a glutamate residue (E62[LC3A/B]) instead, which cannot serve as a H-bond donor to main chain carbonyl residues in

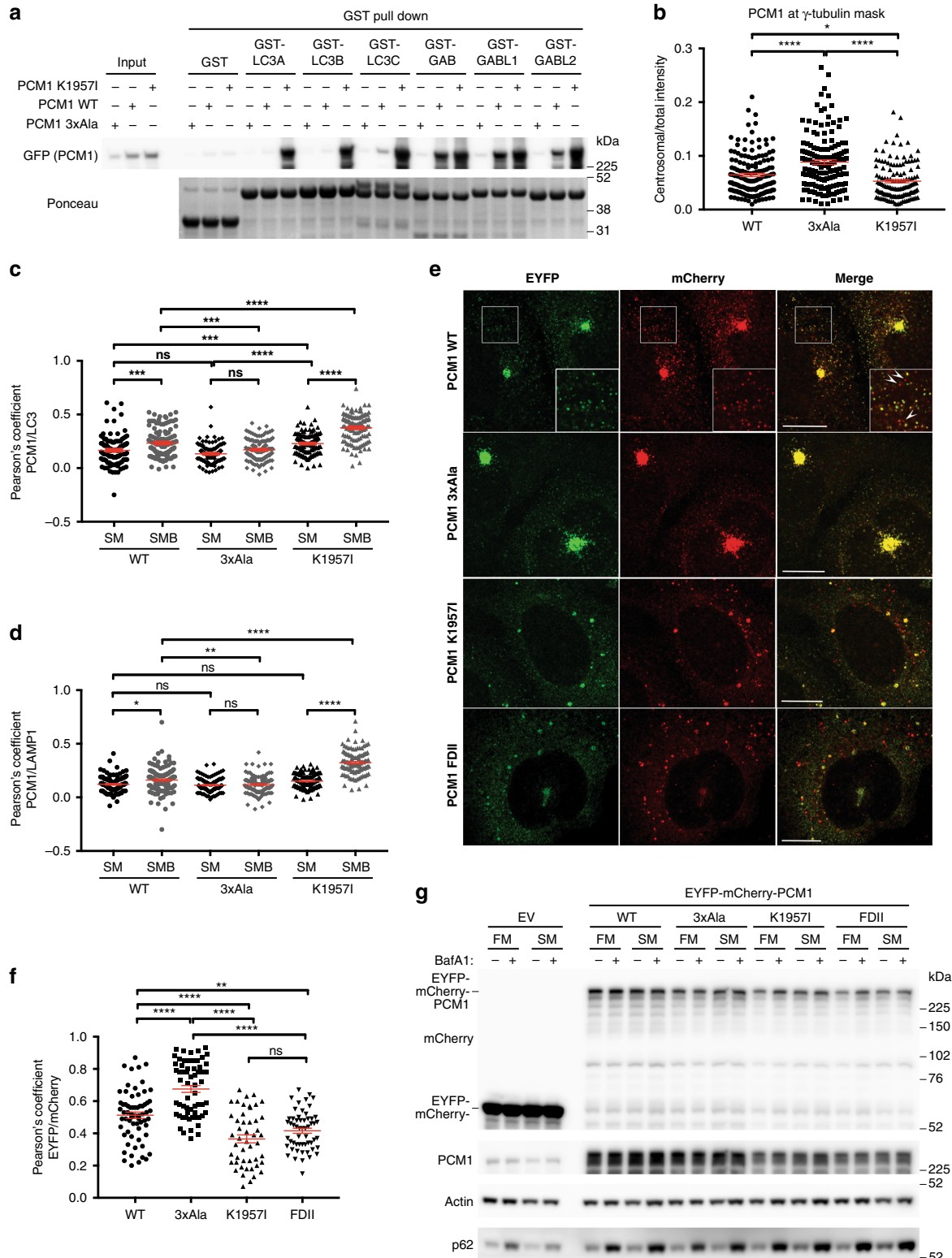

position X$_{7,8}$. Thus, mutation of GABARAP Q59 to glutamate may affect PCM1 and ULK1 complex binding.

In Fig. 1d, asterisks denote the residues that we identified (based on our structures or previously published structures[30–33,39]) and mutated in GABARAP and LC3B, making GABARAP more LC3B-like and vice versa. Changes in PCM1, ULK1, ATG13, FIP200, and p62 binding were determined by GFP-TRAP IP experiments (Fig. 6a–f and Supplementary Fig. 6) and/or BLI

affinity measurements (Fig. 6g). Q59E$^{GAB}$ mutation only slightly affected PCM1 and ULK1 complex binding to GABARAP in IP experiments (Fig. 6a–e and Supplementary Fig. 6a, e), whereas F60L$^{GAB}$ reduced interaction with PCM1 by ~90%. For ULK1, ATG13, and FIP200, we observed the strongest reduction in binding for K24Q/Y25H/Q59E/F60L$^{GAB}$.

Interestingly, the LC3B mutation E62Q$^{LC3B}$, corresponding to Q59E$^{GAB}$ in GABARAP, led to an increase in both PCM1 and

**Fig. 5** Increased LC3 binding alters pericentriolar material 1 (PCM1) dynamics in cells. **a** GST pulldown of HEK293A cells expressing indicated GFP-PCM1 constructs and immunoblot. **b** Quantification of GFP signal intensities at the centrosome (centriole and pericentriolar material marked by γ-tubulin) normalized to total GFP signal intensities of whole cells. Each measurement represents one cell; $n = 4$ independent experiments. **c**, **d** Quantification of GFP-PCM1 colocalization with LC3 (**c**) and LAMP1 (**d**); HEK293A cells expressing indicated GFP-PCM1 constructs starved for 2 h in Earle's balanced salt solution (EBSS) in the absence (SM) or presence of bafilomycin A1 (BafA1) (SMB), fixed and labeled with anti-LC3 and anti-LAMP1. See also Supplementary Fig. 5c. Pearson's coefficient was measured in 30 cells per condition per independent experiment; $n = 3$ independent experiments. **e** HEK293 cells expressing indicated EYFP-mCherry-PCM1 constructs starved for 6 h in EBSS and fixed for confocal microscopy. Scale bars represent 10 μm. **f** Quantification of EYFP colocalization with mCherry. HEK293 cells expressing indicated EYFP-mCherry-PCM1 constructs (shown in **e**) starved for 6 h in EBSS. Pearson's coefficient was measured in at least 50 cells per condition; $n = 3$ independent experiments. **g** HEK293 cells expressing indicated EYFP-mCherry-PCM1 constructs or empty vector (EV) were incubated in full medium (FM) or EBSS (SM) for 6 h in the presence or absence of BafA1 prior to immunoblotting. Statistical analysis using one-way analysis of variance (ANOVA) test; mean ± s.e.m.; $****p \leq 0.0001$; $***p \leq 0.001$; $**p \leq 0.01$; $*p \leq 0.05$; ns not significant

ULK1 complex association (Fig. 6b, d, f and Supplementary Fig. 6b, f). The strongest increase in binding of PCM1 and the ULK1 complex was observed for Q26K/H27Y/E62Q$^{LC3B}$. The presence of the fourth mutation L63F$^{LC3B}$ (corresponding to F60L$^{GAB}$) did not further improve binding. Affinity measurements further verified reduced binding of GABARAP K24Q/Y25H/Q59E/F60L$^{GAB}$ and increased binding of LC3B Q26K/H27Y/E62Q$^{LC3B}$ to PCM1 and ULK1 complex LIR peptides (Fig. 6g). Moreover, we could detect changes (two- to eight-fold) in binding of PCM1 and ULK1 complex LIR peptides to GABARAPs carrying single mutations of K24Q$^{GAB}$, Y25H$^{GAB}$, R28K$^{GAB}$, and Q59E$^{GAB}$, as well as to LC3Bs with single mutations of Q26K$^{LC3B}$, H27Y$^{LC3B}$, K30R$^{LC3B}$, and E62Q$^{LC3B}$. Triple mutations of K24Q/Y25H/R28K$^{GAB}$ in GABARAP and Q26K/H27Y/K30R$^{LC3B}$ in LC3B also lead to stronger changes in PCM1 and ULK1 LIR association. Single mutations of GABARAP residues L55V$^{GAB}$, F62K$^{GAB}$, and L63I$^{GAB}$, which form hydrophobic interactions with the C-terminal region of the PCM1 and ULK1 complex LIRs (Fig. 2f), had no effect or only a small effect on ULK1 LIR binding.

In our IP experiments, F60L$^{GAB}$ interacted significantly less with p62 and this was rescued primarily by the K24Q$^{GAB}$ mutation (Supplementary Fig. 6c, e). As previously reported, mutation of R10E/R11H$^{LC3B}$ reduced LC3B binding to p62[14,39] (Supplementary Fig. 6d, f). Interestingly, the presence of the E62Q$^{LC3B}$ mutation partially rescued p62 binding in this mutant. No increase in PCM1 and ULK1 complex binding to LC3B was observed when mutating both R10E/R11H$^{LC3B}$ and E62Q$^{LC3B}$ (Fig. 6b–f and Supplementary Fig. 6b, f). In GABARAP, the corresponding E8R/H9R$^{GAB}$ mutation (Fig. 6a–e and Supplementary Fig. 6a, c, e) increased PCM1, ULK1 complex, and p62 binding, suggesting that R10/R11$^{LC3A/B}$ and R16/K17$^{LC3C}$ are non-conserved residues critical for PCM1, ULK1 complex, and p62 binding by LC3 subfamily proteins.

**Rendering GABARAP more LC3B-like impairs NDP52 degradation.** Two recent studies showed that ATG8 proteins are dispensable for autophagosome formation, but essential for autophagic flux[12,13]. Loss of all six ATG8 proteins results in p62 accumulation and defects in PINK1/Parkin-dependent clearance of mitochondria in HeLa cells. GABARAP proteins are crucial for autophagosome–lysosome fusion and the major drivers of PINK1/Parkin mitophagy and starvation induced autophagy.

To determine whether altering the binding specificity of GABARAP and LC3B affects their ability to rescue autophagic flux, i.e. p62 degradation, we generated stable inducible ATG8 hexa knockout (KO) HeLa[12] cell lines expressing either MYC-tagged WT GABARAP, GABARAP Q59E/F60L, GABARAP K24Q/Y25H/Q59E/F60L, WT LC3B, or LC3B Q26K/H27Y/E62Q. p62 degradation was significantly rescued by all five constructs (Fig. 7a, b). GABARAP WT, LC3B WT, and LC3B Q26K/H27Y/E62Q exhibited the strongest and comparable

activities. Mutations rendering GABARAP more LC3B-like notably affected its ability to rescue p62 turnover.

Interestingly, the levels of the autophagy receptor NDP52 were also elevated in hexa KO HeLa cells compared to control (Fig. 7a, c and Supplementary Fig. 7a). Both WT GABARAP and LC3 Q26K/H27Y/E62Q rescued NDP52 turnover, whereas no significant activity was observed for LC3B, GABARAP Q59E/F60L, and GABARAP K24Q/Y25H/Q59E/F60L.

Expression of MYC-tagged proteins and formation of puncta in response to starvation was also verified by immunofluorescence staining (Supplementary Fig. 7b). MYC-GABARAP K24Q/Y25H/Q59E/F60L formed fewer and smaller spots, suggesting that impaired ULK complex binding may affect its recruitment to forming autophagosomes.

In summary, rendering GABARAP more LC3B-like impaired its function in autophagic flux, while a more GABARAP-like LC3B,rescued selective degradation of NDP52.

## Discussion

LIR-dependent interactions of autophagy adaptors and receptors with the LDS of mammalian ATG8 proteins are crucial in both non-selective and selective autophagy[21,40]. The fact that many ATG8-interacting proteins preferentially bind to a specific subfamily indicates distinct, non-overlapping functions of LC3 and GABARAP subfamily proteins. Structural studies have been critical for understanding LIR-dependent interactions. However, the majority of analyses have focused on the core LIR and preceding N-terminal region. A potential role of residues C-terminal to the core LIR motif has been, apart from few exceptions (e.g., FYCO1[30–32], ALFY[33], AnkG/AnkB/FAM134B[34]), largely not investigated.

In this study, we used N- and C-terminally extended LIR motifs in X-ray crystallography, affinity measurements, and peptide array binding experiments to elucidate molecular determinants mediating selective binding of the ULK1 complex and PCM1 to GABARAP subfamily proteins. Regarding the region N-terminal of the LIR our data is in line with previous findings[21]. Acidic residues in $X_{-1}$ and $X_{-2}$ are crucial for ULK1 and PCM1 LIR binding, as substitution (Figs. 3b, 4c, d) or removal of the whole N-terminal region of ULK1 (Supplementary Fig. 3b, c) reduced binding to all ATG8 proteins. More strikingly, we identified position $X_2$ within the core LIR motif as well as the adjacent C-terminal region ($X_{4–10}$) as key sites critical for selective binding of ULK1 complex members and PCM1 to GABARAPs.

There is still a lack of in-depth understanding of the molecular determinants in GABARAP and LC3 subfamily proteins mediating selective binding to LIRs. Our data show that binding of these C-terminally extended LIRs involves several non-conserved, GABARAP subfamily-specific residues, which are part of the LDS. Although the LDS is highly conserved among all ATG8 orthologs, these subfamily-specific residues create differences in

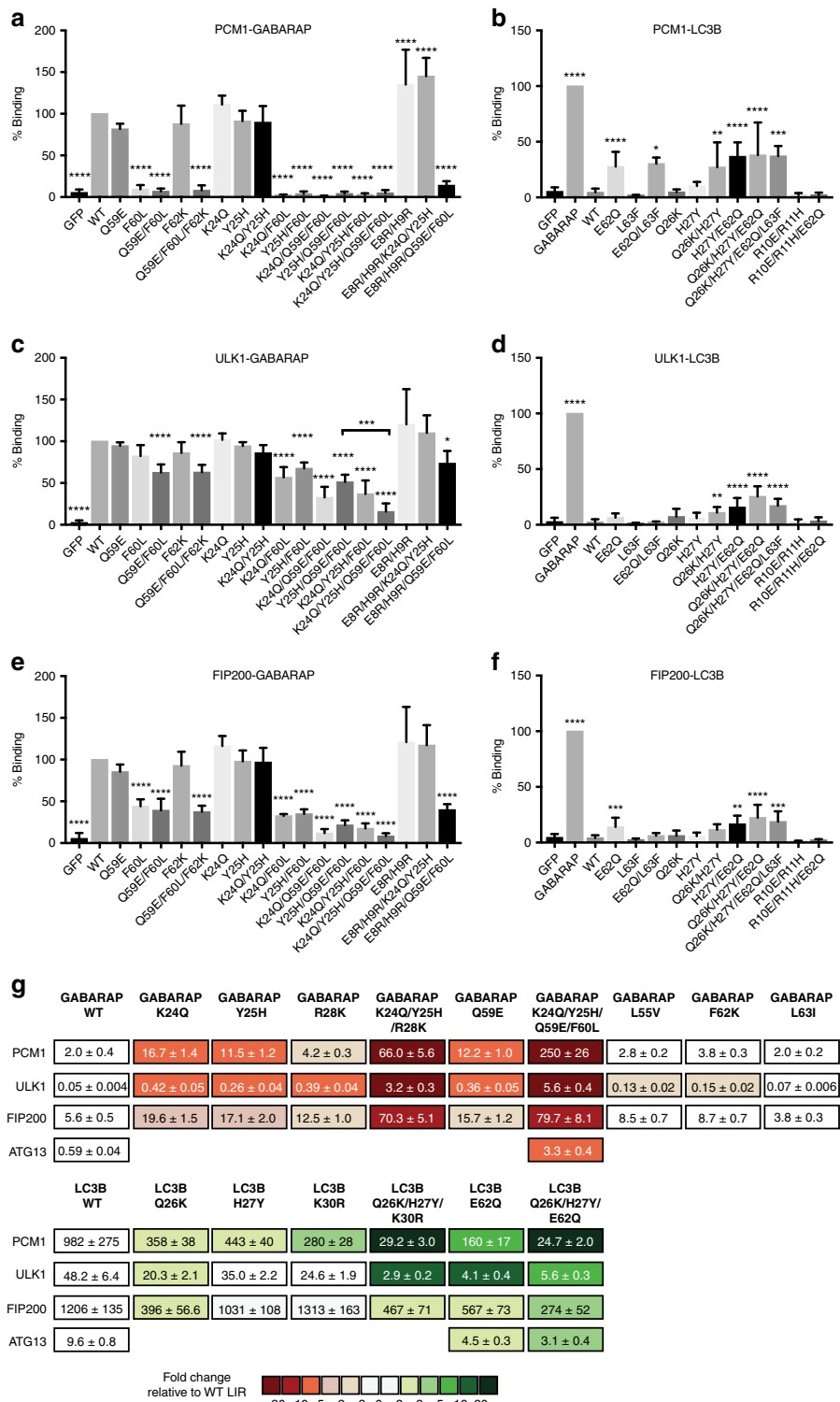

**Fig. 6** Non-conserved GABARAP residues are key to selective LC3-interacting region (LIR) binding. **a**, **c**, **e** Quantification of endogenous pericentriolar material 1 (PCM1) (**a**), ULK1 (**c**), and FIP200 (**e**) binding to indicated GFP-GABARAP constructs expressed and immunoprecipitated by GFP-TRAP from HEK293A cells. Representative immunoblot of GFP-TRAP immunoprecipitation (IP) experiments is shown in Supplementary Fig. 6e. **b**, **d**, **f** Quantification of endogenous PCM1 (**b**), ULK1 (**d**), and FIP200 (**f**) binding to indicated GFP-LC3B constructs expressed and immunoprecipitated by GFP-TRAP from HEK293A cells. Representative immunoblot of GFP-TRAP IP experiments is shown in Supplementary Fig. 6f. Statistical analysis using one-way analysis of variance (ANOVA) test; mean ± s.d.; data from at least three independent experiments. ****$p \leq 0.0001$; ***$p \leq 0.001$; **$p \leq 0.01$; *$p \leq 0.05$. **g** Affinities ($K_d$ values) of LIR peptides to indicated ATG8 proteins measured by bio-layer interferometry (BLI). Color code represents fold changes relative to wild-type (WT) GABARAP and LC3B, respectively (data are mean ± s.d., $n = 2$–5)

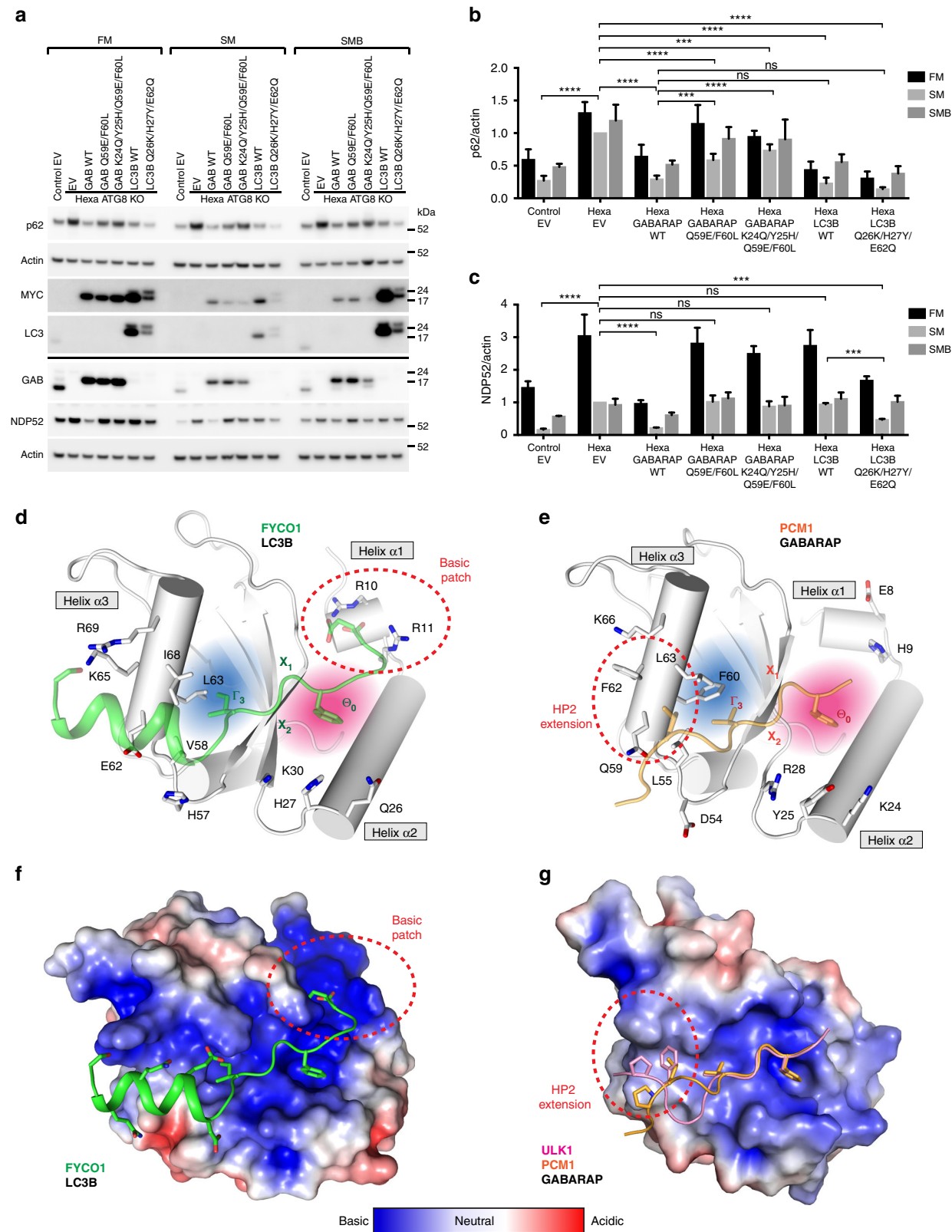

the binding properties of the LDS, which ultimately direct the binding preference of the ULK1 complex and PCM1 LIR motifs towards GABARAPs and against LC3s (Fig. 7d–g).

In contrast to GABARAPs (E8/H9$^{GAB}$ or D8/H9$^{GABL1/2}$), LC3s exhibit basic residues (R10/R11$^{LC3A/B}$ or R16/K17$^{LC3C}$) on helix α1 at the edge of HP1 (Fig. 1d and Fig. 7d–g), which are crucial for FYCO1[30–32] and p62[21,39] (Supplementary Fig. 6d, f) binding, as they form electrostatic interactions with acidic residues in position X$_{-2}$ to X$_{-4}$. Mutation of E8R/H9R$^{GAB}$ in GABARAP enhanced ULK1 complex and PCM1 binding (Fig. 6a, c, e and Supplementary Fig. 6a,

**Fig. 7** Rendering GABARAP more LC3B-like impairs NDP52 degradation. **a** Control (wild-type) and hexa ATG8 CRISPR knockout (KO) HeLa cell lines stably expressing indicated MYC-ATG8 constructs or empty vector (EV) in full medium, Earle's balanced salt solution (EBSS) (SM) or EBSS + bafilomycin A1 (BafA1) (SMB) for 7 h prior to lysis and immunoblot with indicated antibodies. Expression of MYC-ATG8 constructs was induced by 1 μg/ml doxycycline for 6 days. Note, immunoblot of non-induced cells showing equal p62 and NDP52 protein levels in the reconstituted hexa ATG8 CRISPR KO HeLa cell lines is included as Supplementary Fig. 7a. **b, c** Quantification of p62 (**b**) and NDP52 (**c**) protein levels (normalized to actin) shown in **a**. Statistical analysis of (SM) using one-way analysis of variance (ANOVA) test; mean ±s.d.; data from four (p62) and three (NDP52) independent experiments. ****$p \leq 0.0001$; ***$p \leq 0.001$; **$p \leq 0.01$; *$p \leq 0.05$; ns, not significant. **d** Structure of FYCO1 LIR motif bound to LC3B (PDB ID 5WRD). Non-conserved LIR docking site (LDS) residues of LC3B are displayed in white sticks. LC3B is displayed in white cartoon and the FYCO1 LIR in green transparent cartoon. Hydrophobic pocket (HP) 1 and HP2 are indicated in pink and blue, respectively. **e** Structure of PCM1 LIR motif bound to GABARAP ($P2_12_12_1$). Non-conserved LDS residues of GABARAP are displayed in white sticks. GABARAP is displayed in white cartoon and the PCM1 LIR motif is shown in orange transparent cartoon. **f** Surface electrostatic potential of FYCO1:LC3B structure in the same orientation as shown in **d**. **g** Surface electrostatic potential of PCM1:GABARAP ($P2_12_12_1$) structure with ULK1 LIR superimposed (pink cartoon). **f, g** Projections shown are −5 (red) and +5 (blue) kT/e using pymol apbs plugin with red and blue representing negative and positive potential, respectively. Red dashed lines encircle basic patch in LC3B (**d**, **f**) and extension of HP2 in GABARAP (**e**, **f**)

c, e), whereas the R10E/R11H$^{LC3B}$ mutation prevented increased ULK1 complex and PCM1 binding to the LC3B E62Q mutant (Fig. 6b, d, f and Supplementary Fig. 6b, d, f). Moreover, introducing the N-terminal region of the FYCO1 LIR (acidic residues only in $X_{-3}$ and $X_{-4}$) (Fig. 4a), as well as mutating the acidic residues in $X_{-1}$ and $X_{-2}$ (Fig. 4c) in the PCM1 LIR, weakened GABARAP subfamily binding to PCM1. This suggests that acidic residues in $X_{-3}$ and $X_{-4}$ promote specificity towards LC3 rather than GABARAP subfamily proteins. Consistent with previous results[21,30–32,39], R10/R11$^{LC3A/B}$ (R16/K17$^{LC3C}$) are non-conserved residues critical for LC3 subfamily binding to the ULK1 complex and PCM1.

There is growing evidence that position $X_7$ plays a vital role in aiding ATG8 binding. A recent study showed that a short amphipathic α-helix (C-helix) C-terminal to the canonical LIR motifs of AnkG, AnkB, and FAM134B facilitates super-strong binding to ATG8 proteins[34]. Similarly, a shorter C-helix is also critical for FYCO1 LIR binding to LC3A/B[30,31] (Fig. 7d–f). A glutamate residue (in position $X_7$) at the beginning of all these C-helices forms salt bridges with R67$^{GAB}$ or R70$^{LC3B}$, which are strictly conserved among all ATG8 proteins. Negatively charged glutamate or aspartate residues are highly prevalent in position $X_7$[31] and involvement of the C-terminal region in stabilizing ATG8 binding may not be a unique feature of a few LIR sequences, but instead a more general mechanism facilitating LIR binding.

Within the C-terminal region of the PCM1, ATG13, and ULK1 LIR motifs, residues $X_{5–10}$ are important for GABARAP binding. Position $X_7$ of the PCM1, ATG13, and ULK1 LIR is not an acidic residue and we did not detect C-helices. Instead, Q59$^{GAB}$ and L55/F62/L63$^{GAB}$ form hydrogen bonds and hydrophobic interactions with main chain carbonyl and side chain residues in position $X_{5–10}$ (Fig. 2f and Fig. 7e–g). In GABARAPs these aromatic and aliphatic aa on helix α3 expand slightly the hydrophobic surface of HP2, whereas in LC3 subfamily proteins (especially LC3A/B), the edge of HP2 is less hydrophobic and even charged due to V58/K65/I68$^{LC3A/B}$ (Figs. 1d, 7d–g and Supplementary Fig. 2c, d). The single mutations of L55V$^{GAB}$, F62K$^{GAB}$, and L63I$^{GAB}$ (Fig. 6g), producing small differences in the hydrophobic properties of HP2, had little or no effect, suggesting that contributions from these residues are not sufficient to alter LIR specificities. In contrast, Q59E$^{GAB}$ mutation in GABARAP decreased (Fig. 6g) and E62Q$^{LC3B}$ mutation in LC3B increased binding of ULK1 complex and PCM1 LIRs (Fig. 6b–g and Supplementary Fig. 6b, f). Q59E$^{GAB}$ is only conserved in GABARAPs and LC3C and the corresponding residue in LC3A/B (E62$^{LC3A/B}$) is not able to function as hydrogen bond donor, indicating that E62$^{LC3A/B}$ hinders binding to LC3A/B and is contributing to selectivity. Notably, both Q59$^{GAB}$ and E62$^{LC3B}$ stabilize binding of the AnkG/AnkB C-helix in position $X_{12}$[34]. The LDS of GABARAPs can accommodate both C-helices and extended polypeptides, which indicates more plasticity and

dynamics in the interaction of the LDS with the C-terminal region of LIRs than anticipated.

Alongside positions $X_{7–10}$, a proline in position $X_4$ inhibits both ULK1 and FIP200 LIR binding to LC3 subfamily proteins (Fig. 3, Supplementary Fig. 4b, g). Due to the cyclic nature of the side chain, a proline in this position might induce geometric constraints on the LIR polypeptide chain unfavorable for LC3A/B interaction. Interestingly, E1959T$^{PCM1}$ mutation in the PCM1 LIR (Fig. 4c) weakened only LC3 subfamily binding. In the mutational peptide array scan of the FIP200 LIR region, only hydrophobic, aromatic, and acidic aa (V, L, I, G, Y, W, F, E, D, and C) at $X_4$ improved LC3B binding, whereas mainly polar and basic aa did not (T, S, M, K, R, Q, N, P, H, and A) (Supplementary Fig. 4b). This suggests that apart from proline other residues in $X_4$ may also hinder LC3 subfamily binding. More structural analyses using C terminally extended LIR motifs will help to understand the underlying mechanism.

A recent study focusing on the core LIR motif defined a GIM, [W/F]-[V/I]-$X_2$-V, exhibiting preferential binding to GABARAP subfamily proteins[25]. Mutation of both $X_1$ and $X_2$ residues modulated the binding specificity of the PLEKHM1 LIR towards different ATG8 proteins[25]. Position $X_2$ in the LIR is not a strong determinant for GABARAP binding[18,41] (Figs. 3a, 4d and Supplementary Fig. 4a) as only few substitutions (glycine, proline) disrupt the interaction. Although ULK1 and PCM1 have a valine at $X_1$, we found that position $X_2$ in the ULK1, FIP200, and PCM1 LIR motifs is hindering binding to LC3s. LIR binding to LC3 proteins is dramatically altered by substitutions of $X_2$: hydrophobic (I, L, V), and aromatic (F) amino acids strongly increase interaction of LC3B with ULK1, FIP200, and PCM1 LIR motifs (Fig. 3b–e and Supplementary Fig. 4b, c, g). Interestingly, the GABARAP-specific LIR of KBTBD6 exhibits an arginine (basic aa similar to lysine) in position $X_2$[25,42]. Moreover, substitution of the $X_2$ residue with arginine renders the AnkG LIR highly selective for GABARAPs[34]. Furthermore, by introducing a lysine in position $X_2$ (as found in the PCM1 LIR), we could effectively impair the binding of the ATG4B, FUNDC1, and FKBP8 LIR motifs to LC3s (Supplementary Fig. 4d–f), and alter the binding specificity of ATG4B towards GABARAPs. On the other hand, substitution of position $X_2$ in PCM1(K1957$^{PCM1}$) with isoleucine resulted in almost 100-fold increase in affinity to LC3B and ~12-fold increase to GABARAP (Fig. 4c). In cells, mutation of K1957I$^{PCM1}$ in PCM1 drastically switches on LC3 binding, enabling strong interaction with all six ATG8 proteins and altering PCM1 localization and dynamics (Fig. 5).

Why is the $X_2$ LIR residue of PCM1 affecting binding to LC3s more than GABARAPs? Both M359$^{ULK1}$ and M446$^{ATG13}$ (position $X_2$ in the ULK1 and ATG13 LIRs) form hydrophobic interactions with Y25$^{GAB}$ and L50$^{GAB}$, whereas the charged and

longer side chain of K1957$^{PCM1}$ only partially engages with the hydrophobic surface of GABARAP (Supplementary Figs. 2d, 7c). In the same way, the hydrophobic surface of LC3 subfamily proteins is too small to accommodate lysine or arginine side chains (Supplementary Fig. 7d). Moreover, lysine/arginine in X$_2$ clash with nearby positively charged R28$^{GAB}$ and K30$^{LC3A/B}$ and are generally not ideal for LIR binding to any ATG8 protein. Structural comparison of LIR-GABARAP complexes (this study and refs. [33,41,43]) shows that the conformation of K24Q$^{GAB}$, Y25H$^{GAB}$, and R28K$^{GAB}$ are well conserved in all structures (Supplementary Fig. 7c). R28$^{GAB}$ is hydrogen bonding with the carbonyl of the LIR residue in position $\Gamma_3$ (except for PCM1 where it forms a salt bridge with the acidic residue in X$_4$). K24$^{GAB}$ stabilizes the aliphatic part of Y25$^{GAB}$, which forms hydrophobic contacts with the aliphatic side chains of LIR residues in X$_2$. The conformations of the equivalent residues in LC3B are also well conserved across LIR-LC3B structures[32,35,44–47] (Supplementary Fig. 7d). In contrast to R28$^{GAB}$ in GABARAP, K30$^{LC3B}$ of LC3B does not form a hydrogen bond with the carbonyl of LIR residues in $\Gamma_3$ and Q26$^{LC3B}$ seems to not stabilize H27$^{LC3B}$, which is uncharged and has aromatic/hydrophobic properties at physiological pH. The single and triple mutations of K24Q$^{GAB}$, Y25H$^{GAB}$, and R28K$^{GAB}$ in GABARAP reduced PCM1, ULK1, and FIP200 LIR binding (Fig. 6g), whereas triple mutation of the corresponding residues in LC3B (Q26K/H27Y/K30R$^{LC3B}$) increased binding. Thus, K24Q$^{GAB}$, Y25H$^{GAB}$, and R28K$^{GAB}$ mediating binding of LIR residues in X$_2$ are critical for the PCM1 and ULK1 complex LIR binding specificity towards GABARAP.

Our study provides new mechanistic insight into the selective binding of LIR motifs to ATG8 proteins, highlighting an important role of the C-terminal region in addition to the core LIR motif and preceding acidic residues. Manipulation of the key sites in either the LIR motif, sequences flanking the LIR motif, or in the ATG8 proteins themselves can alter the binding specificity of proteins, including autophagy adaptors and receptors to ATG8 proteins. Degradation of NDP52, an autophagy receptor, depends on its specific interaction with LC3C[15]. We found that NDP52 degradation also seems to be facilitated by binding to GABARAP, as both GABARAP and LC3B rendered more GABARAP-like rescued NDP52 degradation (Fig. 7a–c). Our findings will be essential for future work to elucidate overlapping and distinct functions of LC3 and GABARAP subfamily proteins as well as the biology of LIR-dependent interactions with autophagy adaptors and receptors.

Finally, the intricate function of PCM1 and CS in autophagy is not well understood. CS are small, electron-dense granules around the centrosome that contain a variety of proteins involved in centrosome assembly/maintenance and ciliogenesis[27,36]. CS traffic along microtubules and deliver centrosomal components to the centrosome in a dynein-dependent manner. Our recent study showed that CS regulate GABARAP-dependent autophagy[26]. GABARAP is a novel component of a distinct CS subpopulation and is stabilized through interaction with PCM1, thereby preventing GABARAP degradation by the proteasome. Altering GABARAP-PCM1 binding may regulate PCM1 function. Loss of ATG8 binding (PCM1 3xAla) increases centrosomal PCM1 (Fig. 5b), suggesting that PCM1 trafficking from the centrosome may be affected. Moreover, both enhanced ATG8 binding (PCM1 K1957I) and LC3 binding (PCM1 FDII) alters CS integrity and causes PCM1 to form large cytosolic granules (Fig. 5e and Supplementary Fig. 5). Interestingly, deletion of the PCM1 C-terminus likewise induces the formation of large cytosolic PCM1 aggregates[48,49]. The C-terminal region inhibits PCM1 self-aggregation, thereby regulating the size of PCM1-positive CS[48]. Strong association of both LC3s and GABARAPs with the C-terminal PCM1 LIR might promote PCM1 self-aggregation and impair proper CS formation and function. Whether ATG8

proteins and/or autophagy regulate PCM1-dependent CS trafficking, assembly, and size needs to be addressed by future research.

In line with previous findings[26,38] PCM1 is not appreciably degraded by autophagy. Interestingly, increased LC3 binding targeted some PCM1 K1957I and PCM1 FDII into autophagosomes for lysosomal degradation (Fig. 5e, f). Whether binding specificity to GABARAPs protects autophagy adaptors like PCM1 from autophagic degradation is an interesting question for future studies. Moreover, it is becoming apparent in recent years that the centrosome, ciliogenesis machinery and autophagy proteins are intimately linked[24,26,38,50]. Understanding how these pathways are co-regulated is a fascinating direction for future research.

## Methods

**Antibodies.** The following primary antibodies were used: horse radish peroxidase (HRP)-conjugated anti-GST (GE Healthcare, #RPN1236, 1:5000); anti-PCM1 (for western blot (WB) Atlas antibodies, #AMAb90565, 1:500; for immunofluorescence (IF), Sigma, #SAB1406228, 1:200, Cell signaling, #5213, 1:250); anti-ULK1 (Santa Cruz, #sc-33182, 1:250); anti-ATG13[51] (1:1000); anti-FIP200 (Bethyl Labs, #A301-536A-1, 1:2000); anti-p62 (Abnova, #H00008878-M0, 1:500); anti-MYC (Cancer Research UK (raised in-house), clone 9E10, 1:500 for WB, 1:250 for IF; Abcam, #ab9106, 1:1000); anti-HA (Covance, clone 16B12, #MMS-101R-500, 1:1000); anti-GFP (Cancer Research UK (raised in-house), clone 3E1, 1:1000; Santa Cruz, #sc-8334, 1:1000); anti-γ-tubulin ascites (Sigma, Clone GTU-88, #T6557, 1:5000); anti-SSX2IP (ThermoFisher, #PA5-18258, 1:250); anti-pericentrin (Abcam, #ab4448, 1:250); anti-LC3 (Abcam, #ab48394, 1:1000 for WB, 1:2500 for IF); anti-LAMP1 (BD Biosciences, #34201A, 1:250); anti-actin (Abcam, #ab8227, 1:1000); anti-GABARAP (Abgent, #AP1821a, 1:250); anti-NDP52/CALCOCO2 (Abcam, #ab68588, 1:1000); anti-mCherry (Abcam, #ab167453, 1:1000).

Secondary antibodies for IF were all from Life Technologies and diluted 1:500: anti-rabbit immunoglobulin G (IgG) Alexa Fluor 488 (#A11034), 555 (#A31772), and 647 (#A31573); anti-mouse IgG Alexa Fluor 488 (#A21202), 555 (#A31570), 647 (#A31571); anti-goat IgG Alexa Fluor 594 (#A21468). HRP-conjugated secondary antibodies used for WB were from GE Healthcare (#NA931 (anti-mouse IgG), #NA934 (anti-rabbit IgG), 1:4000).

**Plasmids.** EGFP-PCM1 (NP_001302436) (pEGFP-C2)[52] was a gift from Takashi Toda (Hiroshima University, Japan). EGFP-PCM1 (pEGFP-C2) 3xAla (D1954A/ F1955A/V1958A) was generated by us previously[26]. EGFP-PCM1 (pEGFP-C2) K1957I and pDNOR221 PCM1 FDII (V1956D/K1957I/V1958I) were generated using Q5 Site-Directed Mutagenesis Kit (New England BioLabs). EYFP-mCherry-PCM1 WT, 3xAla (D1954A/F1955A/V1958A), K1957I, and FDII (V1956D/ K1957I/V1958I) were generated by using the Gateway recombination system from Invitrogen. BP reactions into pDNOR221 and LR reactions into pDEST-EYFP-mCherry were performed according to the manufacturer's instructions. pDEST-EGFP-ATG8 homologs, pDEST-MYC-ATG8 homologs (human), and pDEST-EYFP-mCherry were generated by Terje Johansen (UiT, The Arctic University of Norway, Tromsø)[18,53]. pDEST-EGFP-GABARAP and LC3B mutants were generated by QuickChange Multi Site-Directed Mutagenesis Kit (Agilent) and Q5 Site-Directed Mutagenesis Kit (New England BioLabs).

HA-ULK1 (human ULK1) in pcDNA3.1 was generated by us previously[54]. HA-ULK1 P361D and ULK1 T354E/M391I/P361D (pcDNA3.1) were generated using Q5 Site-Directed Mutagenesis Kit (New England BioLabs).

The sequences of human GABARAP WT, Q59E/F60L or K24Q/Y25/Q59E/F60L and LC3B WT or Q26K/H27Y/E62Q were inserted together with a N-terminal MYC-tag in the pLVX-TetOne-Puro vector (Clontech) using AgeI/BamHI restriction sites.

For recombinant protein expression in bacteria, a modified plasmid of pGEX-6P2 (GE Healthcare) (referred to as pAL) containing an N-terminal glutathione S tag followed by a 3C protease cleavage site was used. BamHI and NotI restriction sites were used for insertion of full-length human GABARAP, GABARAP-L1, LC3A, and LC3C, and EcoRI and NotI restrictions sites for cloning of LC3B and GABARAP-L2. For crystallization, LIR sequences of PCM1 (aa 1951–1964), ATG13 (aa 441–454), and ULK1 (aa 354–366) were N terminally fused with a Gly-Ser linker to full-length GABARAP (GST-GABARAP (pAL)) using NcoI and BamHI sites. pAL-GST-GABARAP and LC3B mutants were generated using Q5 Site-Directed Mutagenesis Kit (New England BioLabs).

All plasmid constructs generated in this study were verified by DNA sequencing.

See Supplementary Table 1 for primers used in this study.

**Cell culture and DNA transfection.** HEK293A cells (provided by Cell Services of the Francis Crick Institute) were grown in a humidified incubator at 37 °C in 10% CO$_2$ in full medium (Dulbecco's modified Eagle's medium supplemented with 10% fetal calf serum and 4 mM L-glutamine). Control and ATG8 hexa KO HeLa cells[12] were a gift from Michael Lazarou (Monash University, Australia). For

reconstitution of ATG8 hexa KO HeLa cell lines with TetOn-inducible MYC-tagged GABARAP and LC3B proteins cells were transfected with pLVX-TetOne-Puro-MYC-GABARAP/LC3B vector DNA using jetPrime transfection reagent (Polyplus-transfection) according to the manufacturer's instructions. Transduced cells were selected with 2 µg/ml puromycin. Following single-cell sorting, protein expression was induced with 1 µg/ml doxycycline for 4–6 days and colonies were screened for MYC-GABARAP/LC3B expression by western blotting. Verified clones were maintained in full medium containing 1 µg/ml puromycin. For rescue experiments, MYC-GABARAP/LC3B expression was induced with 1 µg/ml doxycycline for 6 days. To induce autophagy by starvation, cells were washed two times with (137 mM NaCl, 3.4 mM KCl, 10 mM $Na_2HPO_4$, 1.8 mM $KH_2PO_4$, pH 7.2) (PBSA) and once with Earle's balanced salt solution (EBSS) and then incubated in EBSS for 2 h (7 h for p62 and NDP52 degradation assays). Where indicated, cells were treated with 100 nM Baf A1 (Calbiochem).

Lipofectamine 2000 (Life Technologies) was used for transient transfection of cells according to the manufacturer's instructions. DNA plasmids were used at a concentration of 1 µg/ml of transfection mix. Cells were harvested or fixed after 24–48 h.

**Western blotting.** Cells were lysed in ice-cold TNTE buffer (20 mM Tris-HCl, pH 7.4, 150 mM NaCl, 0.5% w/v Triton X-100, 10% v/v glycerol, 5 mM EDTA) containing EDTA-free Complete Protease Inhibitor cocktail (Roche) and PhosSTOP (Roche). Lysates were cleared by centrifugation and resolved on NuPAGE Bis-Tris 4–12% gels (Life Technologies) followed by transfer onto a PVDF membrane (Millipore). After incubation with primary and secondary antibodies, the blots were developed by chemiluminescence using Immobilon Classico Western HRP substrate (Merck Millipore). Densitometry was performed with ImageJ software. For western blotting of weak signal antibodies, primary antibody was diluted with SignalBoost Immunoreaction Enhancer Kit (Merck Millipore, 407207) and blots were developed with Luminata Crescendo Western HRP substrate (Merck Millipore).

**Immunoprecipitation.** Cells were lysed in ice-cold TNTE buffer (20 mM Tris-HCl, pH 7.4, 150 mM NaCl, 5 mM EDTA, 0.5% w/v Triton X-100, 10% v/v glycerol, 1× Complete protease inhibitor (Roche), 1× PhosSTOP (Roche)). Lysates were pre-cleared with control agarose beads (ChromoTek) for 1 h at 4 °C. GFP-tagged proteins were immunoprecipitated with GFP-TRAP beads (ChromoTek) and MYC-tagged proteins with MYC-Trap beads (ChromoTek) for 2 h at 4 °C. Beads were washed three times with TNTE (w/o PhosSTOP) and bound protein was eluted with 2× Laemmli buffer at 100 °C for 10 min before resolving by sodium dodecyl sulfate-polyacrylamide gel electrophoresis (SDS-PAGE) (4–12% Bis-Tris NuPAGE gels, Life Technologies) and western blotting.

**Peptide array and GST overlay assay.** GST or GST-ATG8 proteins were expressed (from GST-ATG8 (pAL) plasmids) in *Escherichia coli* BL21 (DE3) plysS cells (Agilent, #200132) in LB medium supplemented with 50 µg/ml kanamycin. Expression was induced by the addition of 0.5 mM IPTG at $OD_{600} = 0.6$ and cells were incubated at 25 °C overnight or at 37 °C for 5 h. Harvested cells were lysed in 50 mM Tris-HCl, pH 8.0, 500 mM NaCl, 0.1% Triton X-100, 0.4 mM 4-(2-aminoethyl)-benzenesulfonylfluoride HCl (AEBSF), and 15 µg/ml benzamidine. Fusion protein was batch adsorbed onto Glutathione-Sepharose 4B beads (GE Healthcare). After five washes with wash buffer (50 mM Tris, pH 8.0, 250 mM NaCl, 0.4 mM AEBSF, and 15 µg/ml benzamidine), fusion proteins were eluted in 50 mM Tris, pH 8.0, 2 mM L-glutathione reduced, 0.4 mM AEBSF, and 15 µg/ml benzamidine.

A MultiPep or ResPep SL automated synthesizer (INTAVIS Bioanalytical Instruments AG, Germany) was used for SPOT synthesis of peptide arrays on cellulose membranes[55]. After blocking membranes in TBST with 5% nonfat dry milk, peptide interactions with GST or GST fusion proteins were tested by overlaying the membranes with either 1 µg/ml (mutational peptide array scan) or 2 µg/ml of recombinant protein (all other peptide arrays) for 2 h at room temperature. Membranes were washed in TBST, and bound proteins were detected with HRP-conjugated anti-GST antibody (1:5000, GE Healthcare, RPN1236).

**GST pulldowns.** Per reaction 50 µg GST-tagged protein was bound to 30 µl Glutathione-Sepharose 4B beads for 2 h at 4 °C. After several washes with PBSA, HEK293A lysate was added and beads were incubated at 4 °C overnight. Beads were washed three time with TNTE lysis buffer (20 mM Tris-HCl, pH 7.4, 150 mM NaCl, 5 mM EDTA, 0.5% w/v Triton X-100, 10% v/v glycerol, 1× Complete protease inhibitor (Roche)) before SDS-PAGE and western blotting.

**Protein expression and purification for crystallization.** GST-LIR-GABARAP chimera proteins (pAL) were expressed in *E. coli* Rosetta (DE3) pLysS (Merck, #70956) at 25 °C overnight. Bacteria were harvested by centrifugation and lysed in 50 mM Tris-HCl, pH 8.0, 500 mM NaCl, 0.1% TX-100, 0.5 mM Tris (2-carboxy-yethyl) phosphine (TCEP), 0.4 mM AEBSF, and 15 µg/ml benzamidine. The fusion protein was batch adsorbed onto Glutathione-Sepharose 4B affinity matrix (GE Healthcare) and recovered by cleavage with 3C protease at 4 °C overnight in 50 mM Tris-HCl, pH 8.0, 150 mM NaCl, and 0.5 mM TCEP. The protein was then

further purified by size exclusion chromatography using a Superdex 200 26/60 column (GE Healthcare) equilibrated and run in 25 mM Tris-HCl, pH 8.0, 150 mM NaCl, and 0.5 mM TCEP.

**Crystallization and data processing.** GABARAP chimera proteins were crystallized at 20 °C using the sitting-drop vapor diffusion method with a protein concentration of 10–20 mg/ml. Initial crystallization trial was performed using Qiagen (JCSG core 1–4, AMSO4), Molecular dimension (PACT, Wizard 1–4), and Jena Bioscience (PiPEG). In all cases the drop included 0.5 µl of protein and 0.5 µl of mother liquor. For PCM1[1951–1964]-GABARAP chimeras, crystals grew in 20% w/v PEG 6 K, 0.2 M $CaCl_2$, 0.1 M HEPES, pH 7 ($P2_12_12_1$) and 25% w/v PEG 1500 and 0.1 M SPG pH 8.5 ($P4_3$). For ATG13[441–454]-GABARAP chimera, crystals grew in 0.2 M NaCl, 10% PEG 8000, and 0.1 M Na K phosphate, pH 6.2. For ULK1[354–366]-GABARAP chimera, crystals grew in 0.2 M $MgCl_2$, 0.1 M Na cacodylate, pH 6.5, and 20% PEG1000. Crystals were flash frozen in liquid nitrogen, and X-ray data sets were collected at 100 K at the I04, I04-1 beamline (mx13775) of the Diamond Light Source Synchrotron (Oxford, UK). Data collection and refinement statistics are summarized in Table 1. The data sets were indexed and scaled with xia2[56]. Molecular replacement was achieved by using the atomic coordinates of the peptide-free GABARAP (PDB code: 1GNU) in PHASER[57]. Refinement was carried out using Phenix[58]. Model building was carried out in COOT[59]. Model validation used PROCHECK[60], and figures were prepared using the graphics program PYMOL (http://www.pymol.org).

**BLI assay.** BLI is an optical analytical technique for measuring kinetics of interactions in real time. The biosensor tip surface immobilized with a ligand is incubated with an analyte in solution, resulting in an increase in optical thickness at the biosensor tip and a wavelength shift, which is a direct measure of the change in thickness. BLI analyses of ATG8s binding to immobilized biotinylated LIR peptides were performed using an Octet Red 96 (ForteBio). Biotinylated LIR peptide (50 µg/ml) was immobilized on streptavidin-coated biosensor (SA, ForteBio) and the typical immobilization levels were above 0.3 nm. Ligand-loaded SA biosensors were then incubated with different concentrations of ATG8. All binding experiments were performed in solid-black 96-well plates containing 200 µl of solution (25 mM Tris, pH 7.5, 150 mM NaCl, 0.5 mM TCEP, 0.1% Tween-20, 1 mg/ml bovine serum albumin (BSA)) in each well at 25 °C with an agitation speed of 1000 rpm. Each measurement was repeated two to five times. Dissociation constants of LIR-ATG8 interactions were determined from plotting the increase in BLI response as a function of the protein concentration and fitting using non-linear regression of ForteBio 7.1 data analysis and GraphPad Prism 7 software.

Peptides were synthesized using FMOC (fluorenylmethoxycarbonyl) solid-phase peptide chemistry by the Francis Crick Peptide Chemistry Technology Platform and are listed in Supplementary Table 2.

**Immunostaining and confocal microscopy.** Cells were grown on coverslips, and then fixed with 3% paraformaldehyde in phosphate-buffered saline (PBS) for 20 min before permeabilization with methanol at room temperature for 5 min. Coverslips were then blocked in 5% BSA (Roche) in PBS for 20 min. Coverslips were incubated with primary antibody in 1% BSA in PBS 1 h at room temperature. Coverslips were washed and incubated with secondary antibody in 1% BSA for 1 h. After final washing with PBS and water, coverslips were mounted in mowiol. Images were acquired using a Zeiss LSM 710 confocal microscope (×63 oil-immersion lens), or a Zeiss LSM 880 Airyscan Confocal microscope (×63 oil-immersion lens) and Zeiss ZEN imaging software.

**Quantifications and statistical analysis.** Centrosomal and total intensity of GFP-PCM1 was quantified using ImageJ software. Region of interest was created by manually selecting the expressing cells and γ-tubulin signal was used to identify the centrosomal area in the cells. Pearson's correlation coefficients were calculated using ImageJ software.

The statistical details of all experiments are reported in the figure legends and figures, including statistical analysis performed, error bars, statistical significance, and exact n numbers. Statistics were performed using GraphPad Prism 7 software, as detailed in figure legends.

**Reporting summary.** Further information on research design is available in the Nature Research Reporting Summary linked to this article.

## Data availability
Atomic coordinates and crystallographic structure factors have been deposited in the Protein Data Bank under accession codes PDB 6HYL (PCM1[1951–1964]-GABARAP, $P2_12_12_1$), PDB 6HYM (PCM1[1951–1964]-GABARAP, $P4_3$), PDB 6HYN (ATG13[441–454]-GABARAP), and PDB 6HYO (ULK1[354–366]-GABARAP). The source data underlying Figs. 1e, 3d, 5a, g, 7a and Supplementary Figs. 6e, f, 7a are provided in Supplementary Fig. 8. The source data underlying Figs. 1g, 4c, 5b–d, 5f, 6, 7b, c and Supplementary Figs. 3c, 4c, 6a–d are provided as a Source Data file. All other data that support the findings of this study are available from the corresponding authors upon reasonable request.

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

## Acknowledgements

We thank Michael Lazarou (Monash University) for ATG8 CRISPR KO HeLa cell lines, Takashi Toda (University of Hiroshima) for EGFP-PCM1, Emmanuelle Thinon, Thomas Mercer, Andrew Purkiss (The Francis Crick Institute), Justin Joachim (Kings College London), and Eva Sjøttem (University of Tromsø) for help and advice, the Francis Crick Fermentation Science Technology Platform (STP) for protein expression, the Francis Crick Structural Biology (STP) for technical support, and Diamond Light Source synchrotron for access to beamlines I04 and I04-1(MX13775). This work was supported by the Francis Crick Institute, which receives its core funding from Cancer Research UK (FC001187 and FC001999); the UK Medical Research Council (FC001187 and FC001999); and the Wellcome Trust (FC001187 and FC001999). The work in T.J.'s laboratory was funded by grants from the FRIBIOMED (grant number 214448) and the TOPPFORSK (grant number 249884) programs of the Research Council of Norway, and the Norwegian Cancer Society (grant number 71043-PR-2006-0320).

## Author contributions

M.W. and M.R. performed cell and biochemical studies. S.M., M.W., W.Z. and L.N. performed structural studies. D.J. and N.O'R synthesized peptides and peptide arrays. M.W., S.M., S.A.T. and T.J. wrote the manuscript. All authors discussed the results and commented on the manuscript. M.W., S.M. and S.A.T. supervised the work. M.W. coordinated the project.

## Additional information

**Competing interests:** The authors declare no competing interests.

