## [Peer Review File · Nature Communications]

Reviewers' comments:

Reviewer #1 (Remarks to the Author):

In this manuscript, Wirth et al. examine the binding specificity of ATG8 interacting proteins for distinct ATG8 family members. In contrast to previous studies on LC3 and GABARAP interactions the authors investigated the impact of residues preceding and succeeding the ATG8 interacting motif (AIM, also known as LC3 interacting region or LIR) core of several interactors with ATG8 subfamily binding preference. Comprehensive biochemical, biophysical and structural analyses revealed a critical contribution of residues C-terminal to the LIR core of the GABARAP-selective ATG8 interactors PCM1, ULK1 and ATG13. In the case of the centriolar protein PCM1, the authors elegantly demonstrate that the binding specificity can be switched from GABARAP to LC3B in vitro and that altered ATG8 specificity had an impact on the integrity and dynamics of centriolar satellites. Through an extensive binding analysis in cells, the authors provided convincing evidence that residues outside the conserved LIR docking site of GABARAP are involved its binding to PCM and ULK1. Consistent with the role of GABARAPs in maintaining autophagy flux the authors showed that the introduction of LC3B-mimicking mutations in GABARAP impaired the degradation of autophagy receptors. Together, the study of Wirth and colleagues provided new mechanistic insights into the properties of human LC3 and GABARAP proteins with important implications for understanding their differential roles in cells. In addition, I am convinced that this work will serve as valid resource for the community and blue print for other studies on ATG8 interacting proteins. While the mutational analysis could of course be extended to the analysis of other autophagy phenotypes (besides turnover of p62 and NDP52), I strongly believe that this should be part of a separate effort and therefore recommend to publish this study without further delay.

Reviewer #2 (Remarks to the Author):

The manuscript by Wirth et al. performed detailed structural and biochemical analyses of the interactions between mammalian ATG8 homologs and several LIR sequences that derived from the centriolar satellite protein PCM1 and ULK1 complex components (ULK1, ATG13, FIP200). Structural data indicate that some GABARAP-subfamily specific residues such as L55, Q59, F62, and L63 are involved in the interaction with the residues at the C-terminal region adjacent to the core LIR motif (positions X5~X10), whereas biochemical data indicate that position X2 in the core LIR motif and position X4 determine the specific binding to GABARAP-subfamily but not to LC3 subfamily. Based on these data, the authors designed GABARAP and LC3B mutants whose specificity were exchanged and analyzed the effect of these mutations on PCM1 function and autophagy receptor degradation in cells.

Overall, structural and biochemical studies have been extensively performed with high quality, and the obtained information will further deepen our understanding of the binding specificity of Atg8 homologs. However, the new findings obtained by structural analyses are not supported by biochemical analyses and vice versa, which make it difficult to derive more general rules that determine the specificity of Atg8 homologs from the enormous data. Thus far, many structural and biochemical studies have been reported for the interaction between Atg8-family proteins and binding motifs. This study at the present form is just a new example of such studies and will not appeal to the wide readers of Nature Communications. It is important for the authors to derive general rules from the enormous biochemical data and explain the rules, at least to some extent, by structural data.

Major points

1) Structural studies clearly show that L55, F62, and L63 of GABARAP form hydrophobic interactions and Q59 of GABARAP forms hydrogen bonds with the C-terminal region (positions X5-X10) of ULK1, ATG13, and PCM1 LIRs. Since these residues are conserved in GABARAP subfamily and partially conserved in LC3C, but not in LC3A and LC3B, these residues seem to be the determinants of different specificity between GABARAP and LC3 subfamilies. However, biochemical data did not support that at all. Binding assays used in Figure 6a-f are not suitable for detecting small changes in affinity, and significant decrease in affinity could be detected only for F60L-containing mutants for GABARAP. Study the binding affinity of L55, Q59, F62, and L63 single mutants of GABARAP with LIRs using BLI.

2) By mutational analyses, positions X-3, X2 and X4 within ULK1 LIR were proposed to regulate selective binding to GABARAP subfamily proteins. These conclusions are important, but not supported by structural data. Since there exist many structural data of Atg8-family proteins in the protein data bank, compare the structural difference between GABARAP and LC3 subfamilies and explain structurally why these positions could regulate the specificity.

3) In the experiments of PCM1 localization, the authors suggested that activation of LC3 binding may target PCM1 K1957I to autophagosomes (in line 299, page 12) and concluded that changing the binding specificity of the LIR can alter the dynamics of PCM1 in cells (in line 308, page 12). However, K1957I mutation enhances the affinity not only with LC3 subfamily but also with GABARAP subfamily (Figure 4c; about 12-fold increase in affinity with GABARAP subfamily) and we cannot judge whether the mutational effects observed in cells are due to the affinity change with LC3 subfamily or with GABARAP subfamily (or both). In order to claim that, the authors must use a mutant that has increased affinity with LC3 subfamily while keeping the same affinity with GABARAP subfamily.

4) In page 14, line 348, the authors summarized that they identified four non-conserved subfamily-specific residues of GABARAP (K24/Y25/Q59/F60) and three of LC3B (Q26/H27/E62), which are critical in regulating selective binding. However, it is not clear whether K24/Y25/Q59 in GABARAP and Q26/H27 in LC3 are actually important because each single mutation did not significantly affect the affinity (Figure 6). The data using multiple mutations are not enough for supporting the important role of these residues because multiple mutations often affect the overall folding of each protein. Study the binding affinity of each single mutant using BLI.

5) In line 366, page 15, the authors suggested that LC3B Q26K/H27Y/E62Q may be more active than LC3B WT due to its lower expression level. But this speculation is too biased. In order to claim that, lower the expression level of LC3B WT or increase the expression level of the mutant and compare the activity under the same expression level. From Figure 7a, it is more reasonable to conclude that LC3B is as active as GABARAP for p62 turnover and the accumulation of p62 observed for GABARAP mutant is not due to its specificity change to LC3 subfamily but only due to the reduced activity of GABARAP. p62 shows similar affinity with both GABARAP and LC3 subfamilies whereas NDP52 shows preference to LC3C (and GABARAP subfamily), which seem to be consistent with the data in Figure 7a. Reconsider the interpretation of the Figure 7a data carefully. In the same reason, the heading that "Autophagy receptor degradation is impaired by rendering GABARAP more LC3B-like" in line 352 may be applicable to NDP52, but not to p62, and thus should be reconsidered.

Minor points

1) In page 11, line 277, describe which residue is mutated for GFP-PCM1(3xAla).

2) In Figure 3b, why the intensity of the dots in the top line are not constant?

3) Show the crystal packing around the PCM1 LIR in both P212121 and P43 in order to show whether the packing affects the interaction or not.

4) Show the crystal packing around ATG13 LIR A and B in order to show that which conformation is less affected by crystal packing (and thus is more reliable conformation).

Reviewer #3 (Remarks to the Author):

The role of Atg8 family members in mammalian system has been the focus of multiple studies in recent years. However, the exact details governing specificity for each family member in the interaction with autophagic receptors or autophagic adaptors remain rather elusive. In the present study, Wirth et al. characterize the interaction between GABARAP, an Atg8 family member and the centriolar satellite protein PCM1. The author mapped the residues important for GABARAPs binding in the LIR motif and in the near C terminal region as well as residues in the GABARAPs docking

site. Moreover, a more detailed analysis within the Atg8 family members has been performed leading to the identification of specific residues within GABARAP that are responsible for the interaction with PCM1. Changing the specificity of the interaction by mutating the LIR motif enhances PCM1 binding to all ATG8s. These mutations affected PCM1 localization to the centrosome resulting in the recruitment of more PCM1 to autophagosomes. Accordingly, the degradation of p62 and NDP52 was dependent on GABARAP and not on LC3.

Overall, this study addresses an interesting question in the field of autophagy regarding the molecular details of Atg8 family members specific interaction. The first part of this study in which the authors characterize the molecular details of GABARAP-PCM1 interaction is well conducted and is mostly convincing. However, the data dealing with the effect of the different mutants on overall autophagy is rather limited and are not fully convincing. Providing additional more detailed experimental analysis on the effect of the different mutants on autophagy is essential to solidify the proposed model.

Additional comments

Fig 3b- The authors should relate to the fact that WT-ULK binding is not equal in the control line.

Fig 5c- The data presented in this figure are not clear. A more detailed analysis of the inreaction/localization of PCM1 wt and mutants should be performed. According to the authors statement – PCM1 should not be localized within the lysosomal lumen as it seems not to be degraded. This could be resolved for example by protease protection assay or by immune-EM. Moreover, the WB analysis of the different mutants in the presence or absence of Baf A should be added to the figure. Using another autophagy marker is more appropriate in order to establish elevated PCM1 localization to forming autophagosomes.

Fig 7a- The analysis of this experiment is not convincing. In this experiment LC3 acts similar to GABARAP. Calibrating the expression levels of the different MYC-ATG8s is needed to accurately interpret this experiment. Moreover, the effect of Baf A should be examined.

Thank you very much for the review of our paper entitled “Molecular determinants within and C-terminal to the core LIR motif mediate selective binding of autophagy adaptors and receptors” (Ref: NCOMMS-18-33840-T).

We thank all three reviewers for excellent comments and criticisms that have helped us to improve our manuscript. In the revised manuscript all text changes are indicated in red.

Detailed response to reviewers comments:

Reviewer #1 (Remarks to the Author):

In this manuscript, Wirth et al. examine the binding specificity of ATG8 interacting proteins for distinct ATG8 family members. In contrast to previous studies on LC3 and GABARAP interactions the authors investigated the impact of residues preceding and succeeding the ATG8 interacting motif (AIM, also known as LC3 interacting region or LIR) core of several interactors with ATG8 subfamily binding preference. Comprehensive biochemical, biophysical and structural analyses revealed a critical contribution of residues C-terminal to the LIR core of the GABARAP-selective ATG8 interactors PCM1, ULK1 and ATG13. In the case of the centriolar protein PCM1, the authors elegantly demonstrate that the binding specificity can be switched from GABARAP to LC3B in vitro and that altered ATG8 specificity had an impact on the integrity and dynamics of centriolar satellites. Through an extensive binding analysis in cells, the authors provided convincing evidence that residues outside the conserved LIR docking site of GABARAP are involved its binding to PCM and ULK1. Consistent with the role of GABARAPs in maintaining autophagy flux the authors showed that the introduction of LC3B-mimicking mutations in GABARAP impaired the degradation of autophagy receptors. Together, the study of Wirth and colleagues provided new mechanistic insights into the properties of human LC3 and GABARAP proteins with important implications for understanding their differential roles in cells. In addition, I am convinced that this work will serve as valid resource for the community and blue print for other studies on ATG8 interacting proteins. While the mutational analysis could of course be extended to the analysis of other autophagy phenotypes (besides turnover of p62 and NDP52), I strongly believe that this should be part of a separate effort and therefore recommend to publish this study without further delay.

Answer: We thank the reviewer for this strong and encouraging support of our work.

Reviewer #2 (Remarks to the Author):

The manuscript by Wirth et al. performed detailed structural and biochemical analyses of the interactions between mammalian ATG8 homologs and several LIR sequences that derived from the centriolar satellite protein PCM1 and ULK1 complex components (ULK1, ATG13, FIP200). Structural data indicate that some GABARAP-subfamily specific residues such as L55, Q59, F62, and L63 are involved in the interaction with the residues at the C-terminal region adjacent to the core LIR motif (positions X5~X10), whereas biochemical data indicate that position X2 in the core LIR motif and position X4 determine the specific binding to GABARAP-subfamily but not to LC3 subfamily. Based on these data, the authors designed GABARAP and LC3B mutants whose specificity were exchanged and analyzed the effect of these mutations on PCM1 function and autophagy receptor degradation in cells. Overall, structural and biochemical studies have been extensively performed with high quality, and the obtained information will further deepen our understanding of the binding specificity of Atg8 homologs. However, the new findings obtained by structural analyses are not

supported by biochemical analyses and vice versa, which make it difficult to derive more general rules that determine the specificity of Atg8 homologs from the enormous data. Thus far, many structural and biochemical studies have been reported for the interaction between Atg8-family proteins and binding motifs. This study at the present form is just a new example of such studies and will not appeal to the wide readers of Nature Communications. It is important for the authors to derive general rules from the enormous biochemical data and explain the rules, at least to some extent, by structural data.

We thank the reviewer for their appreciation of the high quality of our structural and biochemical data. We have performed additional experiments as requested and used data from the literature to start to derive a more general view of the complex code presented by the LIRs and LDS. We feel our contribution to the existing knowledge is substantial in particular our data which highlights the importance of position X₂ within the core LIR motif and the C-terminal region adjacent to the core LIR motif.

Major points

1) Structural studies clearly show that L55, F62, and L63 of GABARAP form hydrophobic interactions and Q59 of GABARAP forms hydrogen bonds with the C-terminal region (positions X5-X10) of ULK1, ATG13, and PCM1 LIRs. Since these residues are conserved in GABARAP subfamily and partially conserved in LC3C, but not in LC3A and LC3B, these residues seem to be the determinants of different specificity between GABARAP and LC3 subfamilies. However, biochemical data did not support that at all. Binding assays used in Figure 6a-f are not suitable for detecting small changes in affinity, and significant decrease in affinity could be detected only for F60L-containing mutants for GABARAP. Study the binding affinity of L55, Q59, F62, and L63 single mutants of GABARAP with LIRs using BLI.

Answer: We cloned and purified mutant GABARAP proteins carrying single mutations of L55, Q59, F62 and L63 for BLI measurements. The determined affinities of all single mutant proteins are shown in Fig. 6g. We did not observe any (or only minor) changes in PCM1, ULK1, and FIP200 LIR binding for the L55V, F62K and L63I mutations. Leucine and isoleucine have very similar hydrophobic properties and the L63I mutation therefore did not have any effect. The L55V and F62K mutations slightly reduce the hydrophobicity of the edge of HP2 and had minor effects only on ULK1 LIR binding.

However, changes in affinity were observed for the GABARAP Q59E mutation, resulting in a three- to six-fold reduction of binding to PCM1, ULK1 and FIP200 LIR peptides. Thus, GABARAP Q59 forming hydrogen bonds with carbonyl residues in position X_{7,8} of the PCM1 and ULK1 complex LIR motifs is clearly contributing to LIR binding.

Due to the multivalent nature of the interaction of LIR motifs with the ATG8 LIR docking site (LDS), single mutations of LDS residues, particularly when it comes those inducing small alterations in hydrophobic interactions, appear to be not sufficient to alter LIR specificities.

Nonetheless, our structural analysis shows L55, F62 and L63 form hydrophobic contacts with residues of the PCM1 and ULK1 LIR motifs and we think these non-conserved GABARAP residues together with Q59 nevertheless contribute to binding of the C-terminal region adjacent to the core LIR motif and provide flexibility of the GABARAP LIR docking site to accommodate a broad range of LIR motifs.

2) By mutational analyses, positions X-3, X2 and X4 within ULK1 LIR were proposed to regulate selective binding to GABARAP subfamily proteins. These conclusions are important, but not supported by structural data. Since there exist many structural data of Atg8-family proteins in the protein data bank, compare the structural difference between GABARAP and LC3 subfamilies and explain structurally why these positions could regulate the specificity.

Answer: Regarding position X-3 the T354E mutation promotes LC3B binding very likely due to charge-charge interactions with the non-conserved, basic residues R10 and R11 on helix

$\alpha 1$ in LC3B. As outlined in our discussion (page 17 and Fig. 7d, f) our data indicates that acidic residues in position X-2 to X-4 promote LC3 binding and explains the effect of the T354E mutation in position X-3 of the ULK1 LIR. The NMR structure of the p62 LIR bound to LC3B (PDB code 2K6Q, unpublished data) shows multiple conformations of the N-terminal region, which interacts with LC3B through charge-charge interactions with preceding acidic residues (X-3 to X-1) and basic residues, such as R10 and R11. Due to the flexibility of this region, these interactions are difficult to detect in crystal structures and may require NMR analyses. Regarding position X2 we have added a structural comparison of LIR-GABARAP and LIR-LC3B complex structures. We included these comparisons as Supplementary Figure 7c and d and explain them in the discussion on page 19-20. Moreover, we also analysed the role of non-conserved residues of GABARAP (K24, Y25, R28) and LC3B (Q26, H27, K30), which are involved in binding of the LIR residue in position X2, using BLI affinity measurements (Fig. 6g).

Structural comparison of LIR-GABARAP complexes (Supplementary Fig. 7c) shows that the conformation of K24Q^{GAB}, Y25H^{GAB} and R28K^{GAB} are well conserved. R28^{GAB} is hydrogen bonding with the carbonyl of the LIR residue in position Γ_3 (except for PCM1 where it forms a salt bridge with the acidic residue in X₄). K24^{GAB} stabilizes the aliphatic part of Y25^{GAB}, which forms hydrophobic contacts with the aliphatic side chains of LIR residues in X₂. The conformations of the equivalent residues in LC3B are also well conserved across LIR-LC3B structures (Supplementary Fig. 7d). In contrast to R28^{GAB} in GABARAP, K30^{LC3B} of LC3B does not form a hydrogen bond with the carbonyl of LIR residues in Γ_3 and Q26^{LC3B} seems to not stabilize H27^{LC3B}, which is uncharged and also acts like an aromatic/hydrophobic residue at physiological pH. In the BLI affinity measurements, single and triple mutations of K24Q^{GAB}, Y25H^{GAB} and R28K^{GAB} in GABARAP reduced PCM1, ULK1 and FIP200 LIR binding (Fig. 6g), whereas triple mutation of the corresponding residues in LC3B (Q26K/H27Y/K30R^{LC3B}) increased binding. Due to these structural differences and the changes in affinity, we conclude that the interactions mediated by K24Q^{GAB}, Y25H^{GAB} and R28K^{GAB} are critical for the PCM1 and ULK1 complex LIR binding specificities towards GABARAPs.

Unfortunately, our structural comparison did not allow us to explain the role of the LIR residue in position X₄. At the moment only the ULK1 LIR-GABARAP structure (this study) and the ATG14 LIR-GABARAP structure (Birgisdottir et al. Autophagy 2019) are available to compare binding of LIR motifs with a proline in position X₄. Moreover, in the majority of LIR-ATG8 structures very short LIR sequences were used and the C-terminal region is absent or too short to draw reliable conclusions from them. We hope that this study will also promote change in the research community towards using longer LIR sequences for structural analysis, which will further help to explain the role of position X₄.

3) In the experiments of PCM1 localization, the authors suggested that activation of LC3 binding may target PCM1 K1957I to autophagosomes (in line 299, page 12) and concluded that changing the binding specificity of the LIR can alter the dynamics of PCM1 in cells (in line 308, page 12). However, K1957I mutation enhances the affinity not only with LC3 subfamily but also with GABARAP subfamily (Figure 4c; about 12-fold increase in affinity with GABARAP subfamily) and we cannot judge whether the mutational effects observed in cells are due to the affinity change with LC3 subfamily or with GABARAP subfamily (or both). In order to claim that, the authors must use a mutant that has increased affinity with LC3 subfamily while keeping the same affinity with GABARAP subfamily.

Answer: Our BLI affinity measurements (Fig. 4c) showed that replacing the PCM1 core LIR motif (FVKV) by the FYCO1 core LIR motif (FDII) strongly increases LC3 subfamily binding without altering the binding affinity to GABARAPs. Therefore, to study the dynamics of PCM1 in cells, we generated EYFP-mCherry-tagged PCM1 WT, 3xAla, K1957I and FDII mutant constructs and analysed whether increased ATG8 binding (PCM1 KI) or LC3 binding (PCM1 FDII) results in targeting of PCM1 to the lysosome by autophagy. Formation of red only puncta (representing autolysosomal PCM1) and reduced EYFP/mCherry colocalization was detected for both PCM1 K1957I and PCM1 FDII mutants (Fig. 5e,f), indicating PCM1 targeting to

autophagosomes and degradation by lysosomes was due to increased LC3 binding. In contrast, very little PCM1 WT and hardly any PCM1 3xAla were degraded by autophagy. Moreover, the tandem-tagged PCM1 FDII mutant also formed large punctate structures in the cytosol (as observed for YFP-mCherry-PCM1 K1957I and GFP-PCM1 K1957I described in Fig. 5 e and Fig. S5). Our data therefore suggests that the observed mutational effects are rather due to higher affinity to LC3 subfamily members than increased GABARAP binding.

4) In page 14, line 348, the authors summarized that they identified four non-conserved subfamily-specific residues of GABARAP (K24/Y25/Q59/F60) and three of LC3B (Q26/H27/E62), which are critical in regulating selective binding. However, it is not clear whether K24/Y25/Q59 in GABARAP and Q26/H27 in LC3 are actually important because each single mutation did not significantly affect the affinity (Figure 6). The data using multiple mutations are not enough for supporting the important role of these residues because multiple mutations often affect the overall folding of each protein. Study the binding affinity of each single mutant using BLI.

Answer: We generated and performed BLI affinity measurements using GABARAP proteins carrying single mutations of K24, Y25, Q59 as well as Q26 and H27 single mutant LC3B (Fig. 6g). We detected two- to eight-fold changes in binding of PCM1, ULK1 and FIP200 LIR peptides to GABARAPs carrying single mutations of K24Q^{GAB}, Y25H^{GAB}, and Q59E^{GAB}, as well as to LC3Bs with single mutations of Q26K^{LC3B} and H27Y^{LC3B}. These changes in affinity further support the important role of these non-conserved residues in mediating LIR binding and specificity.

5) In line 366, page 15, the authors suggested that LC3B Q26K/H27Y/E62Q may be more active than LC3B WT due to its lower expression level. But this speculation is too biased. In order to claim that, lower the expression level of LC3B WT or increase the expression level of the mutant and compare the activity under the same expression level. From Figure 7a, it is more reasonable to conclude that LC3B is as active as GABARAP for p62 turnover and the accumulation of p62 observed for GABARAP mutant is not due to its specificity change to LC3 subfamily but only due to the reduced activity of GABARAP. p62 shows similar affinity with both GABARAP and LC3 subfamilies whereas NDP52 shows preference to LC3C (and GABARAP subfamily), which seem to be consistent with the data in Figure 7a. Reconsider the interpretation of the Figure 7a data carefully. In the same reason, the heading that "Autophagy receptor degradation is impaired by rendering GABARAP more LC3B-like" in line 352 may be applicable to NDP52, but not to p62, and thus should be reconsidered.

Answer: We agree with the reviewer that due to the differences in expression levels of LC3B WT and LC3B Q26K/H27Y/E62Q their activities cannot be easily compared and we therefore revised the manuscript as suggested by the reviewer.

Minor points

1) In page 11, line 277, describe which residue is mutated for GFP-PCM1(3xAla).

Answer: Details on the mutated residues [PCM1 D1954A/F1955A/V1958A (PCM1 3xAla)] were added to the manuscript.

2) In Figure 3b, why the intensity of the dots in the top line are not constant?

Answer: The lower intensity is likely to be a technical issue with peptide spotting or western blotting. We included a mutational peptide array analysis of GST-LC3A binding to the ULK1 LIR motif (Fig. S3d). The intensity of the WT LIR peptide spots are more even than in Fig. 3b for GST-LC3B. In both arrays mutation of position X₃, X₂ and X₄ increased GST-LC3A/B binding, indicating an important role of these residues for mediating binding specificity.

3) Show the crystal packing around the PCM1 LIR in both P212121 and P43 in order to show whether the packing affects the interaction or not.

Answer: We included the crystal packing around the PCM1 LIR structures in Supplementary Figure 1f, g. There is no crystal contact in this area which would affect the interaction.

4) Show the crystal packing around ATG13 LIR A and B in order to show that which conformation is less affected by crystal packing (and thus is more reliable conformation).

Answer: We included the crystal packing around the ATG13 LIR structure in Supplementary Figure 1h. There is no crystal contact in this area and none of the alternate conformations are affected by the crystal packing.

Reviewer #3 (Remarks to the Author):

The role of Atg8 family members in mammalian system has been the focus of multiple studies in recent years. However, the exact details governing specificity for each family member in the interaction with autophagic receptors or autophagic adaptors remain rather elusive. In the present study, Wirth et al. characterize the interaction between GABARAP, an Atg8 family member and the centriolar satellite protein PCM1. The author mapped the residues important for GABARAPs binding in the LIR motif and in the near C terminal region as well as residues in the GABARAPs docking site. Moreover, a more detailed analysis within the Atg8 family members has been performed leading to the identification of specific residues within GABARAP that are responsible for the interaction with PCM1. Changing the specificity of the interaction by mutating the LIR motif enhances PCM1 binding to all ATG8s. These mutations affected PCM1 localization to the centrosome resulting in the recruitment of more PCM1 to autophagosomes. Accordingly, the degradation of p62 and NDP52 was dependent on GABARAP and not on LC3.

Overall, this study addresses an interesting question in the field of autophagy regarding the molecular details of Atg8 family members specific interaction. The first part of this study in which the authors characterize the molecular details of GABARAP-PCM1 interaction is well conducted and is mostly convincing. However, the data dealing with the effect of the different mutants on overall autophagy is rather limited and are not fully convincing. Providing additional more detailed experimental analysis on the effect of the different mutants on autophagy is essential to solidify the proposed model.

We thank the reviewer for their appreciation of the GABARAP-PCM1 data, and have further improved this to solidify the conclusion we make on the targeting of PCM1 by GABARAPs and LC3 proteins. For a more detailed analysis of the effects of the different GABARAP and LC3B mutants on autophagy we have been investigating autophagosome formation (WIPI2 puncta formation) in the reconstituted ATG8 hexa KO HeLa cells lines. However, we find that more work is indeed needed to fully understand and correctly interpret the observed phenotypes on starvation-induced autophagosome formation. We clearly see that because of the complexity and time-consuming nature of such experiments we are at this stage not able to include a more detailed analysis of other autophagy phenotypes.

Additional comments

Fig 3b- The authors should relate to the fact that WT-ULK binding is not equal in the control line.

Answer: The lower intensity is likely to be a technical issue with peptide spotting or western blotting. We included a mutational peptide array analysis of GST-LC3A binding to the ULK1 LIR motif (Fig. S3d). The intensity of the WT LIR peptide spots are more even than in Fig. 3b for GST-LC3B. In both arrays mutation of position X₃, X₂ and X₄ increased GST-LC3A/B binding, indicating an important role of these residues for mediating binding specificity.

Fig 5c- The data presented in this figure are not clear. A more detailed analysis of the inreaction/localization of PCM1 wt and mutants should be performed. According to the authors statement – PCM1 should not be localized within the lysosomal lumen as it seems not to be degraded. This could be resolved for example by protease protection assay or by immune-EM. Moreover, the WB analysis of the different mutants in the presence or absence of Baf A should be added to the figure. Using another autophagy marker is more appropriate in order to establish elevated PCM1 localization to forming autophagosomes.

Answer: To determine whether PCM1 K1957I (which exhibits strongly increased LC3 and also enhanced GABARAP subfamily binding) is recruited into autophagosomes, we generated tandem tagged (EYFP-mCherry) PCM1 constructs. In the lysosomal lumen EYFP quenches faster than mCherry and red only puncta indicate lysosomal PCM1. Few and hardly any red only puncta were observed for PCM1 WT and PCM1 3xAla, respectively. Both PCM1 K1957I and PCM1 FDII (which exhibits increased affinity to LC3s without altering binding to GABARAPs) formed red only puncta when expressed in HEK293A cells (Fig. 5 e, f). In this sensitive assay we therefore conclude that the localization of the mutants which have increased affinity to LC3s (PCM1 FDII) or both LC3s and GABARAPs (PCM1 K1957I) is altered compared to WT.

Western blot analysis (Fig. 5 g) showed, that neither WT nor mutant EYFP-mCherry-PCM1 levels change in response to BafA1 treatment, and we think, that only a small fraction of PCM1 (K1957I, FDII) is targeted into autophagosomes due to increased LC3 binding.

This is in line with previous findings. Tang et al. (Nature, 2013) showed that the ciliogenesis inhibitor OFD1, but not PCM1, is degraded by autophagy in response to 24 h serum starvation in non-dividing cells. Moreover, our group (Joachim et al., Current Biology, 2017, Fig. 5A and B) recently demonstrated, that endogenous PCM1 is not degraded by autophagy and PCM1 functions as an autophagy adaptor protein. Furthermore, we already established in this study that PCM1-GABARAP positive centriolar satellites colocalize with WIPI2b, GFP-DFCP1, LC3B and p62 at sites of autophagosome formation (Joachim et al., Current Biology, 2017, Fig. 3 and Supplementary Fig. 2).

Fig 7a- The analysis of this experiment is not convincing. In this experiment LC3 acts similar to GABARAP. Calibrating the expression levels of the different MYC-ATG8s is needed to accurately interpret this experiment. Moreover, the effect of Baf A should be examined.

Answer: Unfortunately, we were not able to obtain clones with more even expression of the different MYC-ATG8s. As requested by the reviewer we examined the effect of BafA1 and included it in Fig. 7a-c. Our analysis shows that the activities of GABARAP and LC3B are similar regarding p62 degradation. However, we see significant differences between GABARAP and LC3B regarding NDP52 degradation. By rendering LC3B more GABARAP-like (LC3B Q26K/H27Y/E62Q) we can significantly increase its activity in NDP52 turnover (in both full medium(FM [*]) and starvation medium(SM [***])).

Additional changes and data included in the manuscript.

1. Molecular weight markers were added to all figures showing Western blotting data.
2. In Fig. 1g affinities of the p62 LIR peptide to all six ATG8 proteins was added. The p62 LIR affinity to LC3B (4.5 μ M) is similar to the affinity (3.2 μ M) previously determined by isothermal calorimetry (ITC) using a C-terminally extended p62 LIR peptide (Goode et

- al. Autophagy 2016). We included this data to demonstrate that Kd values determined by BLI are comparable to Kds determined by ITC.
3. To further strengthen the peptide array data shown in Supplementary Figure 3b we also included BLI affinity experiments (Supplementary Figure 3c). In line with the peptide array data, removal of the ULK1 LIR C-terminal region increases affinity to LC3A and LC3B, further supporting the important role of the C-terminal region in mediating binding specificity to GABARAPs.
 4. Supplementary Figure 4c was included to further strengthen the critical role of position X₂ in LIR binding. T704I^{FIP200} mutation in the FIP200 LIR increased binding to LC3s between 19- to 42-fold and to GABARAPs by 11- to 25-fold. This further underlines that LIR position X₂ is in general critical for mediating binding specificity towards ATG8 subfamily proteins.

In summary, our new data outlined in the detailed response to the reviewers and above paragraph further strengthen the key findings of our manuscript:

- Position X₂ within the core LIR motif and the C-terminal region adjacent to the core LIR motif are regulating selective binding to ATG8 subfamily proteins.
- Modulation of the LIR binding specificity of the centriolar satellite protein PCM1 alters PCM1 localization and dynamics in cells.
- Non-conserved, subfamily-specific LIR docking site residues in GABARAPs and LC3s mediate binding-specificity to PCM1 and ULK1 complex LIRs and are critical for selective degradation of NDP52 by autophagy.

REVIEWERS' COMMENTS:

Reviewer #2 (Remarks to the Author):

The authors have addressed all of my concerns.

Reviewer #3 (Remarks to the Author):

The authors successfully addressed the concerns raised in the first round of review and in its present form the manuscript meets the journal scientific merit.